# Distilling to Hybrid Attention Models via KL-Guided Layer Selection

**Yanhong Li**[1,*]   **Songlin Yang**[2,*]   **Shawn Tan**[3]   **Mayank Mishra**[3]
**Rameswar Panda**[3]   **Jiawei Zhou**[4]   **Yoon Kim**[2]
[1]Allen Institute for AI   [2]MIT   [3]MIT-IBM Watson AI Lab   [4]Stony Brook University
yanhongl@allenai.org   yangsl66@mit.edu

## Abstract

Distilling pretrained softmax attention Transformers into more efficient hybrid architectures that interleave softmax and linear attention layers is a promising approach for improving the inference efficiency of LLMs without requiring expensive pretraining from scratch. A critical factor in the conversion process is layer selection, i.e., deciding on which layers to convert to linear attention variants. This paper describes a simple and efficient recipe for layer selection that uses layer importance scores derived from a small amount of training on generic text data. Once the layers have been selected we use a recent pipeline for the distillation process itself (RADLADS; Goldstein et al., 2025), which consists of attention weight transfer, hidden state alignment, KL-based distribution matching, followed by a small amount of finetuning. We find that this approach is more effective than existing approaches for layer selection, including heuristics that uniformly interleave linear attentions based on a fixed ratio, as well as more involved approaches that rely on specialized diagnostic datasets.[1]

## 1 Introduction

Linear attention (Katharopoulos et al., 2020; Peng et al., 2021; Yang et al., 2023, *i.a.*) and state-space models (Gu et al., 2022; Gu & Dao, 2024; Dao & Gu, 2024, *i.a.*) have gained significant traction recently due to their high inference speed and competitive performance. However, most existing pretrained models are still purely based on softmax attention, and pretraining such linear attention models from scratch is resource-intensive. This has motivated the approaches for *cross-architecture* distillation, a process that converts pretrained Transformer checkpoints into more efficient linear attention counterparts (Kasai et al., 2021; Wang et al., 2024; Bick et al., 2025, *i.a.*).

This distillation process involves two key decisions: (1) the student architecture, and (2) the optimal distillation recipe once the architecture has been selected. For the second question, recent work has shown the effectiveness of a multi-stage pipeline over pure continued finetuning approaches (Bick et al., 2025; Goldstein et al., 2025). This pipeline involves an initial stage of per-layer output alignment with an $L_2$ loss, followed by a second stage of end-to-end knowledge distillation. What student architecture to distill to, however, remains open. Prior efforts to distill Transformers into purely subquadratic models have often resulted in performance degradation (Zhang et al., 2024a;b; Mercat et al., 2024). More recently, models incorporating a sliding window attention (SWA) mechanism have shown surprisingly strong results across various benchmarks (Lan et al., 2025;

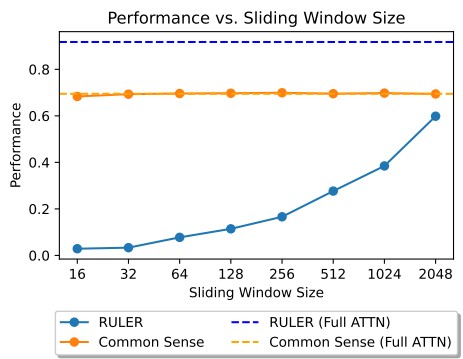

Figure 1: Performance of a sliding-window attention model (distilled from Qwen2.5-3B-Instruct) across different window sizes on RULER and commonsense reasoning tasks.

---

*Equal contribution. Work conducted while YL was a visiting student at MIT.
[1]Code is available at https://github.com/fla-org/hybrid-distillation.

Zhang et al., 2025). However, these evaluations have primarily focused on knowledge-intensive commonsense reasoning tasks, where in-context recall plays a lesser role. Indeed, Figure 1 shows that even a small sliding window of size 16 is sufficient for a distilled SWA model to recover strong performance on such tasks. In contrast, performance on in-context recall benchmarks like RULER (Hsieh et al., 2024) is highly dependent on the sliding window size (Figure 1). This is perhaps unsurprising, as it reflects the well-documented limitations of fixed-state models in in-context recall (Wen et al., 2025; Arora et al., 2024a;b).

A simple yet effective solution is to incorporate a few global (softmax) attention layers, resulting in a hybrid architecture. This approach has been successfully adopted in recent models pretrained from scratch, such as Jamba (Lenz et al., 2025), MiniMax-01 (MiniMax et al., 2025), Falcon-H1 (Zuo et al., 2025), and Qwen3-Next. These models typically interleave global and linear attention layers at a fixed ratio (e.g., one global layer for every three or seven linear layers) (Wang et al., 2025a). Following this trend, some distillation works have also adopted a fixed interleaving strategy (Wang et al., 2024). However, our preliminary experiments show this uniform approach remains suboptimal for in-context recall, presumably due to the fundamental difference between pretraining and distillation. This observation has been recognized in recent work (Gu et al., 2025; Yang et al., 2025; Hoshino et al., 2025), which also explore various criteria for selecting global attention layers.

In this work, we adopt a simple global attention selection criterion based on the distillation KL divergence loss: intuitively, the more critical a global attention layer is, the more it reduces the resulting distillation KL loss. Our experiments demonstrate the effectiveness of our selective hybrid distillation, which achieves strong in-context retrieval performance while maintaining efficiency. Our work paves the path for future work on test-time compute scaling for distilled hybrid models (Paliotta et al., 2025; Wang et al., 2025b), where in-context retrieval remains a key bottleneck (Chaudhry et al., 2025).

## 2 PRELIMINARIES

**Notation.** Let $\mathbf{X} = [\mathbf{x}_1; \ldots; \mathbf{x}_T] \in \mathbb{R}^{T \times d}$ be a sequence of $T$ token embeddings with model width $d$. We use $L$ pre-norm Transformer blocks indexed by $\ell \in \{1, \ldots, L\}$, and $h$ attention heads with per-head width $d_h$ so $d = h \, d_h$. A Transformer block then given by

$$\mathbf{U}^{(\ell)} = \mathbf{X}^{(\ell)} + \text{Mix}^{(\ell)}(\text{LN}(\mathbf{X}^{(\ell)})), \qquad \mathbf{X}^{(\ell+1)} = \mathbf{U}^{(\ell)} + \text{FFN}^{(\ell)}(\text{LN}(\mathbf{U}^{(\ell)})).$$

where $\text{Mix}^{\ell}(\cdot)$ is a sequence mixing operation (i.e., softmax or linear attention) for layer $\ell$. When not essential, we omit LN and residuals for readability. We write $\mathbf{M}$ for the (additive) attention mask, which encodes causality and any positional encoding (e.g., RoPE/Alibi) as standard.

**Softmax attention.** For a single head (we suppress head indices) softmax attention proceeds by computing the query, key and value matrices

$$\mathbf{Q} = \mathbf{X}\mathbf{W}_Q, \quad \mathbf{K} = \mathbf{X}\mathbf{W}_K, \quad \mathbf{V} = \mathbf{X}\mathbf{W}_V,$$

where $\mathbf{W}_Q, \mathbf{W}_K, \mathbf{W}_V \in \mathbb{R}^{d \times d_h}$ are learnable parameters. The output is given by (with mask $\mathbf{M}$)

$$\mathbf{O} = \text{Softmax}\left(\tfrac{1}{\sqrt{d_h}}\mathbf{Q}\mathbf{K}^\top + \mathbf{M}\right)\mathbf{V}, \tag{1}$$

and multi-head concatenates per-head outputs which is transformed by a linear layer $\mathbf{W}_O \in \mathbb{R}^{(h d_h) \times d}$. During autoregressive inference, the same operation admits a recurrent view:

$$\mathbf{o}_t = \sum_{i \leq t} \alpha_{t,i}\,\mathbf{v}_i, \qquad \alpha_{t,i} \propto \exp\left(\tfrac{1}{\sqrt{d_h}}\mathbf{q}_t^\top \mathbf{k}_i\right), \quad \sum_{i \leq t} \alpha_{t,i} = 1. \tag{2}$$

The memory cost of softmax attention grows linearly with respect to sequence length due to the KV cache, which can result in substantial slowdowns as generation length grows due to increasing data movement across the memory hierarchy.

**Linear attention.** Linear attention layers have been proposed to address the above inefficiencies of softmax attention during decoding. While many variants exist, they generally adopt the following recurrent form:

$$\mathbf{o}_t = \mathbf{q}_t^\top \mathbf{S}_t, \quad \mathbf{S}_t = \mathbf{M}_t \mathbf{S}_{t-1} + \mathbf{k}_t \mathbf{v}_t^\top, \tag{3}$$

where $\mathbf{M}_t$ is a data-dependent and time-varying transition matrix that is a function of $\mathbf{x}_t$. Setting $\mathbf{M}_t = \mathrm{diag}(\boldsymbol{\alpha}_t)$ where $\boldsymbol{\alpha}_t \in \mathbb{R}^d$ is a function of $\mathbf{x}_t$ recovers recent gated linear attention (GLA) variants (Yang et al., 2023; Katsch, 2023; Qin et al., 2024; Peng et al., 2024). Alternatively, using $\mathbf{M}_t = \alpha_t(\mathbf{I} - \beta_t \mathbf{k}_t \mathbf{k}_t^\top)$ recovers the (gated) DeltaNet family of models (Schlag et al., 2021; Yang et al., 2024b;a).[2] The structure of $\mathbf{M}_t$ enables efficient parallel training via a chunking mechanism.

Linear attention compresses the entire history into the hidden state matrix $\mathbf{S}_t$ and thus the memory cost is constant with respect to generation length, leading to much more efficient decoding compared to softmax attention. However, this hidden state bottleneck is a fundamental limitation when it comes to crucial capabilities such as performing associative recall over a given context.

**Hybrid attention.** A common strategy for maintaining the capabilities of softmax attention while realizing some of the efficiency benefits of linear attention is to use a hybrid model. This approach partitions the set of layer indices into $\mathcal{S}_{\text{softmax}}$ and $\mathcal{S}_{\text{linear}}$ such that $\mathcal{S}_{\text{softmax}} \cup \mathcal{S}_{\text{linear}} = \{1, \ldots, L\}$. Then the sequence-mixing layer is given by

$$\mathrm{Mix}^{(\ell)} = \begin{cases} \mathrm{SoftmaxAttn}^{(\ell)}, & \ell \in \mathcal{S}_{\text{softmax}}, \\ \mathrm{LinearAttn}^{(\ell)}, & \ell \in \mathcal{S}_{\text{linear}}. \end{cases}$$

Recent works have shown that architectures that use a fixed ratio of linear to softmax attention layers performs well when pretrained from scratch (Lenz et al., 2025; MiniMax et al., 2025). However, such a uniform strategy may be suboptimal for distilling hybrid attention models from pretrained softmax attention models, motivating the present work on layer selection for distillation.

## 3    Layer Selection for Distilling Hybrid Attention

For distilling a pretrained softmax attention LLM into a hybrid attention model, we seek to find a set $\mathcal{L}_{\text{soft}}$ for a given budget $|\mathcal{L}_{\text{soft}}| = K$ such that converting all the other layers into linear attention has minimal performance degradation. Solving this exactly would require a combinatorial search over all possible $K$-sized subsets of $[L]$, which would be intractable. Our key idea is to measure a layer's *marginal utility* by restoring exactly that layer (and only that layer) to softmax in an otherwise all-linear student, then distilling briefly and scoring how much the teacher–student KL improves.

### 3.1    Initial distillation to an all-linear student

We first distill to an all-linear student model, adopting the first two stages of the distillation pipeline from RADLADS (Goldstein et al., 2025). Let $\mathcal{M}_{\text{teacher}}$ be the original teacher model and $\mathcal{M}_{\text{all-linear}}$ be an all-linear student model, where the linear attention parameters are initialized from the teacher's parameters, i.e., $(\mathbf{W}_Q, \mathbf{W}_K, \mathbf{W}_V, \mathbf{W}_O)$. The other parameters of the linear attention layer (in particular the parameters of a linear layer for the data-dependent gating term $\alpha_t$) are initialized randomly. Then distillation proceeds as follows:

**Stage 1: Hidden-state alignment.** For a given token sequence $\boldsymbol{x} = x_1 \ldots x_T$, the attention hidden states from the all-linear student model $\{\mathbf{U}_{\text{all-linear}}^{(\ell)}\}_{\ell \in [l]}$ are trained to match the teacher's hidden states $\{\mathbf{U}_{\text{teacher}}^{(\ell)}\}_{\ell \in [l]}$,

$$\mathcal{L}_{\text{hidden}}(\mathcal{M}_{\text{all-linear}}, \boldsymbol{x}) = \sum_{\ell \in [L]} \frac{1}{T} \big\| \mathbf{U}_{\text{teacher}}^{(\ell)} - \mathbf{U}_{\text{all-linear}}^{(\ell)} \big\|_2^2. \tag{4}$$

Here, RADLADS only trains the parameters of the student's linear attention layer while freezing FFN's parameters. The targets are produced by the teacher model and remain fixed.

**Stage 2: Distribution matching.** In stage 2, RADLADS minimizes a temperature-scaled KL between teacher logits $\boldsymbol{\ell}_{\text{teacher},t} \in \mathbb{R}^V$ and student logits $\boldsymbol{\ell}_{\text{all-linear},t} \in \mathbb{R}^V$ with respect to all student parameters (i.e., including the student's FFN layers)

$$\mathcal{L}_{\text{KL}}(\mathcal{M}_{\text{all-linear}}, \boldsymbol{x}) = \frac{\tau^2}{T} \sum_{t=1}^{T} \mathrm{KL}\Big( \mathrm{Softmax}\Big(\tfrac{\boldsymbol{\ell}_{\text{teacher},t}}{\tau}\Big) \,\Big\|\, \mathrm{Softmax}\Big(\tfrac{\boldsymbol{\ell}_{\text{all-linear},t}}{\tau}\Big) \Big), \tag{5}$$

---

[2]DeltaNet also multiplies the additive term $\mathbf{k}_t \mathbf{v}_t^\top$ with $\beta_t$, which we omit for simplicity.

where $\tau$ smoothing term that provides stronger gradient signal on non-argmax tokens. (The functions $\mathcal{L}_{\text{hidden}}$ and $\mathcal{L}_{\text{KL}}$ are obviously functions of $\mathcal{M}_{\text{teacher}}$ but we omit them for readability.)

Stage 1 uses 100M tokens while stage 2 uses 600M tokens. All subsequent applications of the stagewise pipeline (i.e., in §3.2 and §3.3) use the same number of tokens.[3]

## 3.2 Deriving Layerwise Importance Scores

With the all-linear model $\mathcal{M}_{\text{all-linear}}$ derived from the above process, we now describe our layer selection strategy. Let $\mathcal{M}_{\text{all-linear}}^{(-\ell)}$ be a model derived from $\mathcal{M}_{\text{all-linear}}$ where the $\ell$-th block has been restored back into the $\ell$-th layer of $\mathcal{M}_{\text{teacher}}$. We run stage 1 and stage 2 of the above process again to finetune the student $\mathcal{M}_{\text{all-linear}}^{(-\ell)}$, which now has one softmax attention layer. We define $\mathcal{I}(\ell)$, the layer importance for layer $\ell$, as the KL divergence between and the teacher model, i.e.,

$$\mathcal{I}(\ell) = -\mathbb{E}_{\boldsymbol{x} \sim \mathcal{D}}\big[\mathcal{L}_{\text{KD}}(\mathcal{M}_{\text{all-linear}}^{(-\ell)}, \boldsymbol{x})\big]. \tag{6}$$

Higher $\mathcal{I}(\ell)$ means larger KL reduction (i.e., greater marginal utility under our objective). Because the baseline student and neighbors are fixed, $\mathcal{I}(\ell)$ is hybrid-aware and variant-aware.

## 3.3 Layer Selection and Final Distillation

---

**Algorithm 1** KL-guided Layer Selection for Hybrid Attention Distillation

---

**Require:** Teacher $\mathcal{M}_{\text{teacher}}$; dataset $\mathcal{D}$ (DCLM); temperature $\tau$; target budget $K$
 1: Distill into pure linear attention model $\mathcal{M}_{\text{all-linear}}$ (§3.1)
 2: **for** $\ell = 1$ to $L$ **in parallel do** (§3.2)
 3:     Obtain $\mathcal{M}_{\text{all-linear}}^{(-\ell)}$ by changing $\ell$-th layer of $\mathcal{M}_{\text{all-linear}}$ to $\ell$-th layer of $\mathcal{M}_{\text{teacher}}$
 4:     **Stage 1:** align all linear blocks by $\mathcal{L}_{\text{hid}}$ on $\mathcal{D}$.
 5:     **Stage 2:** distill by $\mathcal{L}_{\text{KL}}$ on $\mathcal{D}$.
 6:     Compute $\mathcal{I}(\ell) = -\mathbb{E}[\mathcal{L}_{\text{KL}}]$ on a held-out slice of $\mathcal{D}$.
 7: **end for**
 8: **Select:** $\mathcal{S}_{\text{softmax}} \leftarrow$ top-$K$ layers by $\mathcal{I}(\ell)$ (§3.3)
 9: **Final hybrid:** instantiate hybrid based on $\mathcal{S}_{\text{softmax}}$ and linear on layers $[L] \setminus \mathcal{S}_{\text{softmax}}$; train with the two-stage distillation pipeline.

---

Given a budget of $K$ softmax attention layers that we can keep, we now take the top-$K$ most important layers and convert the result into linear attention i.e.,

$$\mathcal{S}_{\text{softmax}} = \text{top-K}(\mathcal{I}(\ell)), \quad \mathcal{S}_{\text{linear}} = \{1, \ldots, L\} \setminus \mathcal{S}_{\text{softmax}}.$$

Denoting the above hybrid model with $K$ softmax attention layers as $\mathcal{M}_{\text{hybrid-}K}$ we run a final distillation pipeline by rerunning stages 1 and 2 with this hybrid model. Our full algorithm is given in Algorithm 1.

## 4 Experiments

Having introduced our method, we now present a series of experiments designed to build a comprehensive case for its effectiveness. We begin by establishing why hybrid models are essential for maintaining long-context capabilities (§4.1). We then demonstrate that our KL-guided approach outperforms a wide range of baselines (§4.3).

## 4.1 The Case for Hybrid Models

There has been a flurry of recent work on distilling to pure linear attention models (Chen et al., 2024; Mercat et al., 2024; Zhang et al., 2025; Goldstein et al., 2025; Wang et al., 2024; Yueyu et al., 2025; Lan et al., 2025; Bick et al., 2025). These works generally report that pure linear

---

[3]For our main GA-S2 selector, the final hybrid model reuses the Stage 1-aligned linear attention layers from $\mathcal{M}_{\text{all-linear}}$ and therefore only runs Stage 2 in the last distillation step.

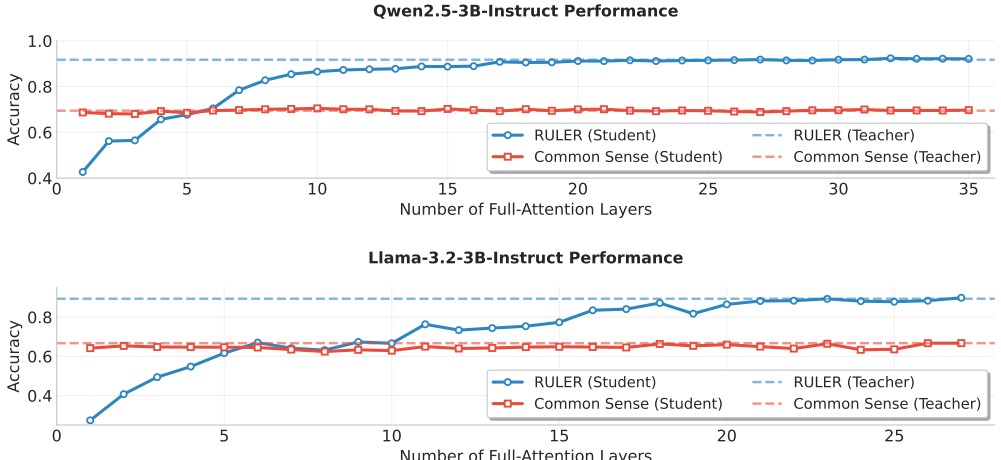

Figure 2: Performance on recall-intensive vs. commonsense tasks as the number of full-attention layers is varied for Qwen2.5-3B-Instruct (top) and Llama-3.2-3B-Instruct (bottom). Recall ability is highly sensitive to the softmax budget, while commonsense reasoning is not.

attention can maintain the performance of pretrained softmax attention baselines with the right distillation process. However, this conclusion is often based on comparing performance on tasks such as MMLU and Commonsense Reasoning, whose context lengths are short; it is unclear the extent to which such pure linear attention models can maintain performance on benchmarks which require understanding and performing recall over longer contexts. To analyze this, we construct a series of hybrid models based on our approach where the number of softmax layers ranges from 1 to $L-1$. We then evaluate these models on RULER (Hsieh et al., 2024), a diagnostic benchmark designed to probe the long-context capabilities of LLMs. We also evaluate these models on short-context commonsense reasoning benchmarks evaluated by previous methods, including PIQA, ARC-Easy, ARC-Challenge, HellaSwag and WinoGrande (we report the average).

The results in Figure 2 reveal a stark dichotomy. Performance on the long-context RULER benchmark is highly sensitive to the number of softmax layers ($K$), growing monotonically and confirming that global context aggregation is critical for in-context retrieval. In contrast, commonsense reasoning performance is almost entirely insensitive to $K$; models with even a single softmax layer achieve near-teacher-level performance, suggesting these local tasks are well-handled by linear attention. Ironically, the efficiency benefits of linear attention are minimal on precisely these short-context tasks. This dichotomy motivates our work: the central challenge in distilling hybrid models is to preserve long-context recall. This requires a method that can judiciously allocate a limited budget of expensive softmax layers to the positions where they are most impactful.

## 4.2 EXPERIMENTAL SETUP

Having established the importance of selection, we now evaluate our KL-guided method against the a suite of baselines.

**Model and data.** We evaluate two 3B-class decoder-only teachers: **Qwen2.5-3B-Instruct** and **Llama-3.2-3B-Instruct**. For each architecture we take the checkpoint's native depth $L$ and report $K$ to match the target softmax:linear ratio. We target four ratios 1:8, 1:3, 1:2, 1:1 (thus $K \in \{4, 9, 12, 18\}$ when $L=36$; if $L$ differs, we use the nearest integer $K$). All selection and distillation runs use the **DCLM** (Li et al., 2025) generic-text mixture. As noted in § 3.1, each instance of stage 1 uses 100M tokens while stage 2 uses 600M tokens.

**Baselines.** We compare our one-swap selector to the baselines below. Each returns a set of $K$ softmax layers and is trained with the same two-stage distillation and token budget as ours (§3.1): (1) **Uniform interleave (UNIFORM).** Pick $K$ layers by evenly spacing them across depth (one roughly

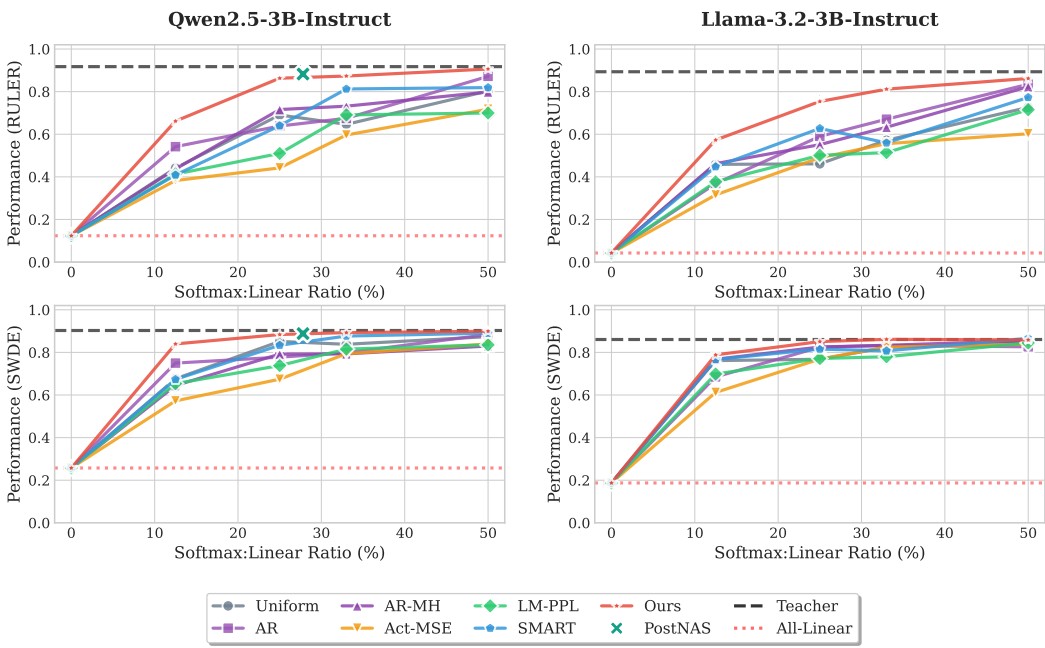

Figure 3: Performance comparison of various layer selection methods on RULER (top) and SWDE (bottom) for distilling Qwen2.5-3B-Instruct (left) and Llama-3.2-3B-Instruct (right) into hybrid GDN-based models. Performance is plotted against the percentage of softmax layers retained. The dashed line indicates the performance of the all-softmax teacher model.

every $\lfloor L/K \rfloor$ blocks), as adopted by Wang et al. (2024). (2) **Task-guided selectors.** AR (Associative Recall): bypass each layer and measure the drop on a synthetic key–value recall task and then rank layer importance by drop in performance (Chaudhry et al., 2025). AR-MH (Associative Recall - Multihop): same as AR but with multi-hop alias chains, which makes the task more difficult. (3) **Model-signal selectors.** ACT-MSE: layer importance is derived from zero-ing out a layer and measuring increase in activation MSE vs. the baseline. LM-PPL: same as Act-MSE, but derived from measuring an increase in LM perplexity on held-out data. (4) **SMART** (Yang et al., 2025). A sensitivity-aware strategy: (i) score each layer by the reduction in teacher–student KL when swapping an global layer into an otherwise linear baseline; (ii) preserve high-score layers near input/output (so-called "terminal preservation"); (iii) choose the rest from near-uniform candidates to maximize total sensitivity. We also compare against **PostNAS** (Gu et al., 2025), a contemporaneous work that uses a more complex search procedure. Their method involves training a once-for-all SuperNet and then using beam search to find the optimal $K$ softmax layers for a specific downstream task. This process is computationally intensive, requiring 50B training tokens, whereas our selection pipeline uses only 5-6B tokens. Fortunately, PostNAS released their selected layers for the Qwen2.5 model. To ensure a fair comparison, we take their publicly released layer set and distill it using our own pipeline and token budget. More baselines descriptions are included in Table 5 in the Appendix A.

## 4.3 MAIN RESULTS

We use gated DeltaNet (GDN) for our linear attention layer and evaluate our proposed layer selection method against the baselines for Qwen2.5-3B-Instruct and Llama-3.2-3B-Instruct teachers. The results on two long-context, recall-intensive benchmarks, RULER and SWDE, are presented in Figure 3. Our central finding is that our selection method consistently and substantially outperforms all other baselines across both models and tasks. This demonstrates the effectiveness of using a brief, KL-divergence-guided distillation to derive model-intrinsic layer importance scores for creating hybrid architectures.

| Teacher | Teacher Performance | 25% softmax | | 33% softmax | |
|---|---|---|---|---|---|
| | | SMART | GA-S2 (Ours) | SMART | GA-S2 (Ours) |
| Qwen2.5-1.5B | 0.8742 | 0.5098 | **0.5408** | 0.6479 | **0.6953** |
| Qwen2.5-7B | 0.9445 | 0.8158 | **0.8584** | 0.8949 | **0.9110** |

Table 1: Experiments on different model sizes on RULER at fixed hybrid budgets.

| | Stage 1 (MSE-based) | | | Stage 2 (KL-based) | | |
|---|---|---|---|---|---|---|
| Model | GR-S1 | GA-S1 | AVG-S1 | GR-S2 | GA-S2 (Ours) | AVG-S2 |
| Llama-3.2-3B-Instruct | 0.4508 | 0.4193 | 0.4233 | 0.4950 | **0.7539** | 0.5580 |
| Qwen2.5-3B-Instruct | 0.4827 | 0.5408 | 0.4933 | 0.8259 | **0.8631** | 0.8205 |

Table 2: Ablation on layer selection strategies for a fixed 25% softmax ratio. We compare Greedy Addition (GA), Greedy Removal (GR), and Averaged (AVG) search using either a Stage-1 (MSE) or Stage-2 (KL) importance metric.

A key advantage of our approach is particularly evident in the low-budget regime, where only a small fraction of layers are kept as full softmax attention. For instance, on RULER with Qwen2.5 at a 12.5% softmax budget (5 softmax layers), GA-S2 reaches 0.662, outperforming the strongest baseline (AR at 0.542) by +0.12 and the standard UNIFORM interleaving strategy (0.441) by +0.22. This pronounced gap at low softmax ratios highlights our method's efficiency in identifying the most critical layers for preserving long-context recall, enabling significant performance gains with minimal computational overhead from expensive attention layers.

As the budget for softmax layers increases, our method continues to maintain a performance advantage, approaching the teacher model's performance more rapidly than competing approaches. For both models, a hybrid with 50% of its layers selected by our method recovers a vast majority of the teacher's performance on these challenging recall tasks. Similar performance trends were observed on other benchmarks, including FDA and SQuADv2; these results are detailed in the Appendix A.

To test whether the gains of KL-guided selection persist beyond the 3B setting, we distill hybrid students from Qwen2.5-1.5B-Instruct and Qwen2.5-7B-Instruct at 25% and 33% softmax budgets (same distillation recipe and data). Table 1 shows GA-S2 consistently outperforms the strongest baseline (SMART) across both scales, with improvements of +0.031/+0.047 (1.5B) and +0.043/+0.016 (7B) at 25%/33%. Full results (including all baselines) are showed in Section G.

## 5 ANALYSIS

In this section, we conduct a series of ablation studies to deconstruct our method (§5.1), understand its architectural sensitivities (§5.2), and validate its practical efficiency (§5.3).

### 5.1 THE IMPORTANCE OF KL AND GREEDY ADDITION STRATEGY

Our proposed layer selection method involves two key design choices: (1) we use the stage-2 (S2) knowledge distillation (KL-based) loss as the importance metric for each layer in the one-swap setting of §3.2, and (2) given these layerwise scores, we select the top-$K$ softmax layers in a greedy addition fashion (GA), i.e., we keep the $K$ layers that yield the largest marginal KL reduction relative to the all-linear baseline. There are natural alternatives: we could use the stage-1 (S1) hidden-state alignment (MSE-based) metric as our layer importance; we could also use a greedy *removal* (GR) search strategy, which starts from an all-softmax model and greedily converts the least important layer to a linear attention layer. It is also possible to average the layer importance rankings from both GA and GR (AVG). Note that our main proposed method corresponds to GA-S2.

The ablation results, presented in Table 2, show that the Stage-2 (KL-based) methods consistently and dramatically outperform their Stage-1 (MSE-based) counterparts, and our greedy addition strategy (GA-S2) is more effective than greedy removal (GR-S2). This suggests that identifying the single most impactful layer to add from an all-linear base is a more robust signal than identifying the least harmful layer to remove. Full layer-wise importance rankings for all selectors are provided in Appendix C.

|  | Llama-3.2-3B | | Qwen2.5-3B | |
|---|---|---|---|---|
| **Ratio** | GDN | GLA | GDN | GLA |
| 12.5% | 0.5731 | 0.5166 | 0.6617 | 0.6074 |
| 25% | 0.7539 | 0.6498 | 0.8631 | 0.6921 |
| 33% | 0.8118 | 0.7967 | 0.8733 | 0.878 |
| 50% | 0.8619 | 0.8487 | 0.9061 | 0.9069 |

Table 3: Final RULER performance using architecture-specific selections.

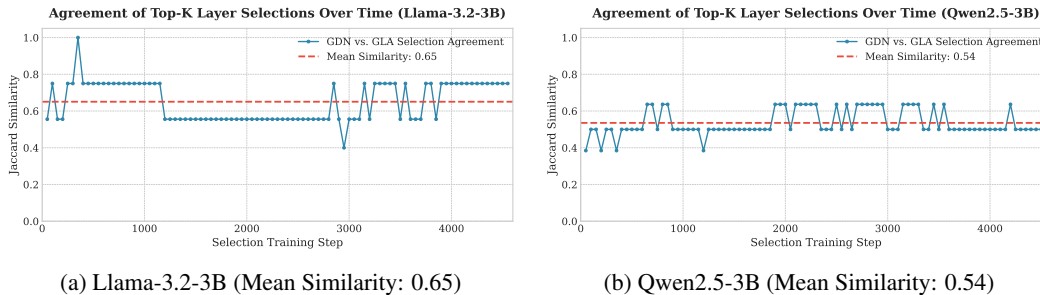

(a) Llama-3.2-3B (Mean Similarity: 0.65)  (b) Qwen2.5-3B (Mean Similarity: 0.54)

Figure 4: Jaccard similarity of top-K layer selections between GDN and GLA variants over the selection pass. Llama shows higher agreement, suggesting its layer importance is less student-dependent.

## 5.2 THE IMPORTANCE OF ARCHITECTURE CONSISTENCY

Our layer selection approach is sensitive to the type of linear attention layer employed. To what extent is this selection approach architecture-agnostic—i.e., is our method simply finding a fixed set of "important layers" in the teacher, or is it adapting its selection to the specific architecture of the student's linear layers? To test this, we run the selection process independently for both GDN and GLA students and analyze the results.

The results in Figure 4 and Table 3 reveal an interesting architectural dependence. At a fixed 25% softmax budget ($K=9$), the GDN and GLA selection trajectories exhibit only moderate overlap: the mean Jaccard similarity is 0.65 for **Llama-3.2-3B-Instruct** and 0.54 for **Qwen2.5-3B-Instruct**, which corresponds to roughly ∼7/9 and ∼6/9 layers overlapping on average (Figure 4). Thus, the two student variants typically disagree on only 1–3 layers, yet these small differences can have an outsized impact on long-context recall. Concretely, in this low-budget regime, the architecture-specific GDN-GDN models substantially outperform the architecture-specific GLA-GLA models on RULER: 0.7539 vs. 0.6498 for Llama and 0.8631 vs. 0.6921 for Qwen (Table 3).

Most surprisingly, when we test transferability by using the GDN-selected layers to distill a GLA student, we achieve strong RULER performance (0.6927 for Llama and 0.8407 for Qwen; Table 4). This result is not only far better than all baselines, but is also significantly better than the score from the specialized GLA-GLA process (0.6498 for Llama and 0.6921 for Qwen). This reveals a key finding: the choice of linear attention variant used during the selection pass acts as a "probe", and some probes are better than others at identifying a robust set of important layers for a given teacher architecture. In particular, using GDN as the probe yields a layer set that transfers well to both GDN and GLA students in the low-budget regime. This demonstrates that our method's strength is not just in specialization, but in its ability to leverage different student architectures to identify the most impactful softmax layers for preserving long-context recall.

## 5.3 HOW MANY TOKENS ARE REALLY NECESSARY FOR LAYER SELECTION?

We used 100M tokens for stage 1 and 600M tokens for stage 2 following the recipe recommended in Goldstein et al. (2025). However, it is possible that the layer selection process could be even more token-efficient. To investigate this, we tracked the top-$K$ layer set chosen by our selector throughout the Stage-2 training process (at a 1:3 softmax ratio for both models). We measured stability over time using rolling-window Jaccard similarity and the size of the intersection between consecutive

| Model | Student | UNIFORM | AR | AR-MH | MSE | PPL | SMART | **Ours** |
|-------|---------|---------|-----|-------|-----|-----|-------|----------|
| Llama | GDN | 0.461 | 0.59 | 0.5512 | 0.4899 | 0.5011 | 0.6274 | **0.7539** |
|       | GLA | 0.4342 | 0.5404 | 0.4838 | 0.4436 | 0.4266 | 0.5851 | **0.6927** |
| Qwen  | GDN | 0.6904 | 0.6385 | 0.716 | 0.4421 | 0.51 | 0.6401 | **0.8631** |
|       | GLA | 0.633 | 0.574 | 0.6628 | 0.3961 | 0.4225 | 0.6007 | **0.8407** |

Table 4: Performance on RULER for GDN- and GLA-based hybrid students at a fixed 25% softmax ratio. For both student variants, the layer set for our method (**Ours**) was selected using a GDN-based process to test for transferability. Note that Llama refers to Llama-3.2-3B-Instruct and Qwen refers to Qwen2.5-3B-Instruct.

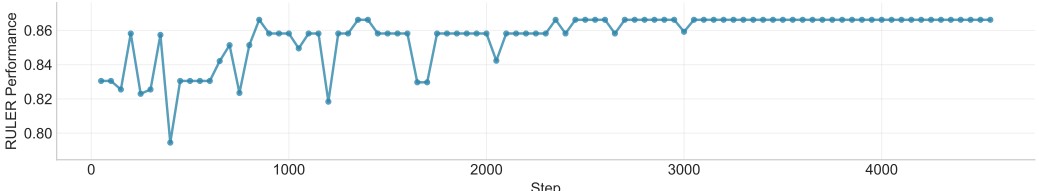

Figure 5: The evolution of RULER performance during the Stage-2 selection process for Qwen2.5-3B-Instruct.

sets (the "backbone"). For both teacher models, we find that the set of selected layers stabilizes long before the full training budget is consumed. A nearly complete "backbone" of $K - 1$ layers is typically identified within the first 25-40% of training. Continuing training beyond this point only refines the choice for the final one or two slots, with a negligible impact on the final model's RULER performance (a difference of less than 0.01 absolute points). This observation suggests that a simple stability-based rule can dramatically improve efficiency. For instance, a conservative early stopping point for our runs would have reduced the token budget for the selection pass by 58–74%. The effectiveness of this early stopping rule is backed by our empirical observation: for Qwen, the RULER performance during Stage-2 stabilizes around step 1500, as shown in Figure 5. For more details, please refer to Appendix B.

## 6 RELATED WORK

In-context recall presents a significant challenge for subquadratic models, a difficulty often attributed to the perplexity gap between them and standard transformers (Arora et al., 2024a). One promising approach to address this is the development of linear attention variants with superior recall capabilities. The seminal work on DeltaNet (Schlag et al., 2021; Yang et al., 2024b) and its successors (Yang et al., 2024a; Siems et al., 2025; Grazzi et al., 2025) has demonstrated great success in this area. Nevertheless, these recurrent approaches are fundamentally limited in associative recall by their fixed-size state (Wen et al., 2025; Arora et al., 2024a). Highlighting the importance of this problem, recent work reveals a connection between in-context recall and test-time scaling performance, arguably making it one of the most critical research directions in efficient sequence model design (Chaudhry et al., 2025). Other notable efforts to improve recall include reading inputs twice (Arora et al., 2024c), dynamic state allocation (Ben-Kish et al., 2025), and dynamic caching for hard-to-memorize items (Nguyen et al., 2025).

Hybrid attention architectures, which combine the complementary strengths of global attention (for accurate retrieval) and linear attention (for fast local processing), can theoretically overcome these state-size limitations (Wen et al., 2025; Arora et al., 2024b). While most hybrid models adopt an inter-layer strategy, interleaving global and linear attention layers (Ren et al., 2025; MiniMax et al., 2025; Lenz et al., 2025), we also note the potential of intra-layer hybridization schemes for efficient time mixing (Irie et al., 2025; Dong et al., 2024; Zuo et al., 2025; Zancato et al., 2024). However, pretraining these linear and hybrid models from scratch is computationally expensive. An effective alternative is to distill a pretrained softmax attention model into a linear attention-based one. This concept was first proposed by Kasai et al. (2021). Subsequent work has emphasized preserving or mimicking the softmax operator during distillation to maintain performance while achieving linear

complexity Peng et al. (2022); Zhang et al. (2024b;a). Research work shows that sliding window attention with window size 64 works well in many benchmarks Lan et al. (2025); Zhang et al. (2025), though we show in this work that such strategies still perform poorly on in-context recall.

In the context of distilling into a hybrid of global and linear attention, a key question has emerged: how to select which global attention patterns to preserve. Some methods rely on downstream benchmark performance to determine importance Gu et al. (2025), while others use speculative decoding as a diagnostic tool to identify redundant attention layers Hoshino et al. (2025). In contrast, our work focuses on a simple strategy using an unsupervised learning loss and provides extensive analysis that goes beyond prior research (Yang et al., 2025).

## 7 CONCLUSION

In this work, we introduced a simple and effective method for selecting which softmax attention layers to retain when distilling a pretrained Transformer into a more efficient hybrid architecture. While our selection process is more efficient than complex search-based alternatives, future work could explore even cheaper proxies for layer importance, potentially derived directly from the teacher model's activations or gradients. Other promising directions include extending this selection framework from the layer level to a more fine-grained, head-level hybridization.

## ACKNOWLEDGMENTS

This work was supported by National Science Foundation under CAREER Award No. 2441872, MIT-IBM Watson AI Lab, and a gift from Jane Street.

## STATEMENT ON LLM USAGE

We acknowledge the use of Large Language Models (LLMs) to assist in the preparation of this manuscript. Specifically, LLMs were utilized to improve grammar and clarity, aid in literature discovery, and generate boilerplate code snippets for our experiments and testing scripts. The authors have carefully reviewed and edited all LLM-generated outputs and take full responsibility for the final content and scientific integrity of this work.

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

# A COMPLETE RESULTS ON RECALL-INTENSIVE BENCHMARKS

| Tag | Selector | Signal / One-Line Procedure |
| --- | --- | --- |
| UNIFORM | Uniform Interleave | Selects layers by evenly interleaving softmax layers at the target ratio. |
| *Task-Guided Search (Heuristic-Based)* | | |
| KV | KV Retrieval | Importance from performance drop on a synthetic key-value dictionary lookup task when a layer is bypassed. |
| AR | Associative Recall | Importance from performance drop on a task to sum the values of prompted keys when a layer is bypassed. |
| AR-MH | Assoc. Recall—Multi-hop | As above, but with alias chains requiring multi-hop reasoning; performance drop defines importance. |
| VT | Variable Tracking | Importance from exact-set accuracy drop on a pointer-chasing task over shuffled assignments. |
| CWE | Common Words Extraction | Importance from set-match accuracy drop on a task to identify the $K$ most frequent words in a long text. |
| ACT-MSE | Activation MSE | Mean-squared error on generic text between the final hidden states of a baseline vs. layer-bypassed model. |
| LM-PPL | LM Perplexity | Measures the increase in perplexity on a held-out corpus when a layer is bypassed. |
| *Greedy Structural Search (Learning-Based)* | | |
| GR–S1 | Greedy Removal (S1) | Starts with all softmax; greedily converts the layer to linear that hurts performance least after brief Stage-1 adaptation. |
| GR–S2 | Greedy Removal (S2) | As above, but using a brief Stage-2 knowledge distillation for adaptation at each step. |
| GA–S1 | Greedy Addition (S1) | Starts with all linear; greedily converts the layer to softmax that helps performance most after brief Stage-1 adaptation. |
| GA–S2 | Greedy Addition (S2) | As above, but using a brief Stage-2 knowledge distillation for adaptation at each step. |
| AVG–S1 | Rank-Avg Greedy (S1) | Averages the layer importance rankings from GR–S1 and GA–S1 before selecting the top-$K$ layers. |
| AVG–S2 | Rank-Avg Greedy (S2) | Averages the layer importance rankings from GR–S2 and GA–S2 before selecting the top-$K$ layers. |

Table 5: Layer-selection baselines and the tags used in figures. Layer bypass means applying an identity residual connection across the block's mixing sublayer.

| Selector | Llama-3.2-3B-Instruct | | | | Qwen2.5-3B-Instruct | | | |
|---|---|---|---|---|---|---|---|---|
| | 12.5% | 25% | 33% | 50% | 12.5% | 25% | 33% | 50% |
| *Heuristic-Based* | | | | | | | | |
| UNIFORM | 0.4586 | 0.4610 | 0.5717 | 0.7262 | 0.4412 | 0.6904 | 0.6464 | 0.8031 |
| KV | 0.2029 | 0.6051 | 0.6626 | 0.7538 | 0.2543 | 0.7539 | 0.7552 | 0.8257 |
| AR | 0.3684 | 0.5900 | 0.6707 | 0.8347 | 0.5417 | 0.6385 | 0.6749 | 0.8717 |
| VT | 0.1839 | 0.2012 | 0.4334 | 0.7538 | 0.2922 | 0.4780 | 0.5359 | 0.7409 |
| CWE | 0.3129 | 0.3579 | 0.6752 | 0.8394 | 0.2900 | 0.4907 | 0.7065 | 0.8444 |
| ACT-MSE | 0.3154 | 0.4899 | 0.5557 | 0.6023 | 0.3827 | 0.4421 | 0.5965 | 0.7175 |
| LM-PPL | 0.3767 | 0.5011 | 0.5138 | 0.7152 | 0.4129 | 0.5100 | 0.6907 | 0.6997 |
| SMART | 0.4476 | 0.6274 | 0.5595 | 0.7728 | 0.4089 | 0.6401 | 0.8126 | 0.8190 |
| AR-MH | 0.4622 | 0.5512 | 0.6329 | 0.8231 | 0.4371 | 0.7160 | 0.7318 | 0.7983 |
| *Learning-Based (S1 - MSE)* | | | | | | | | |
| GR-S1 | 0.2903 | 0.4508 | 0.5214 | 0.6435 | 0.3563 | 0.4827 | 0.6743 | 0.8209 |
| GA-S1 | 0.3092 | 0.4193 | 0.4892 | 0.6569 | 0.3843 | 0.5408 | 0.6657 | 0.7873 |
| AVG-S1 | 0.3108 | 0.4233 | 0.5355 | 0.6390 | 0.3960 | 0.4933 | 0.6441 | 0.8226 |
| *Learning-Based (S2 - KL)* | | | | | | | | |
| GR-S2 | 0.3084 | 0.4950 | 0.6991 | 0.7662 | 0.5804 | 0.8259 | 0.8541 | 0.8869 |
| GA-S2 | 0.5731 | 0.7539 | 0.8118 | 0.8619 | 0.6617 | 0.8631 | 0.8733 | 0.9061 |
| AVG-S2 | 0.4764 | 0.5580 | 0.6786 | 0.8111 | 0.7075 | 0.8205 | 0.8704 | 0.9051 |

Table 6: RULER performance for various layer selection strategies across different softmax ratios, for GDN-based hybrid students . The all-linear (0%) baselines are 0.0427 for Llama-3.2 and 0.1236 for Qwen2.5. The all-softmax teacher scores are 0.8934 and 0.9174, respectively.

| Selector | Llama-3.2-3B-Instruct | | | | Qwen2.5-3B-Instruct | | | |
|---|---|---|---|---|---|---|---|---|
| | 12.5% | 25% | 33% | 50% | 12.5% | 25% | 33% | 50% |
| *Heuristic-Based* | | | | | | | | |
| UNIFORM | 0.3648 | 0.4011 | 0.4728 | 0.6642 | 0.2523 | 0.6334 | 0.3684 | 0.7178 |
| KV | 0.2069 | 0.6461 | 0.6760 | 0.6942 | 0.1370 | 0.6788 | 0.6261 | 0.7096 |
| AR | 0.5000 | 0.7042 | 0.7505 | 0.7051 | 0.4601 | 0.6016 | 0.6379 | 0.7432 |
| VT | 0.1978 | 0.3385 | 0.3648 | 0.6960 | 0.1588 | 0.4183 | 0.4809 | 0.6279 |
| CWE | 0.3149 | 0.3258 | 0.6207 | 0.6779 | 0.0789 | 0.2087 | 0.5345 | 0.6842 |
| ACT-MSE | 0.3058 | 0.5045 | 0.5726 | 0.5672 | 0.1579 | 0.2405 | 0.3385 | 0.5771 |
| LM-PPL | 0.4201 | 0.5000 | 0.5299 | 0.7613 | 0.2523 | 0.2541 | 0.4809 | 0.5508 |
| SMART | 0.3276 | 0.7033 | 0.4374 | 0.7169 | 0.1515 | 0.3448 | 0.7359 | 0.6679 |
| AR-MH | 0.4773 | 0.5572 | 0.6833 | 0.7505 | 0.1588 | 0.6416 | 0.6679 | 0.7069 |
| *Learning-Based (S1 - MSE)* | | | | | | | | |
| GR-S1 | 0.2015 | 0.3548 | 0.5644 | 0.6007 | 0.2677 | 0.4465 | 0.5100 | 0.6697 |
| GA-S1 | 0.2105 | 0.4365 | 0.4746 | 0.5563 | 0.2532 | 0.4247 | 0.5163 | 0.6234 |
| AVG-S1 | 0.1951 | 0.4074 | 0.4628 | 0.6443 | 0.2414 | 0.4165 | 0.5227 | 0.6751 |
| *Learning-Based (S2 - KL)* | | | | | | | | |
| GR-S2 | 0.3303 | 0.5054 | 0.6933 | 0.6633 | 0.3612 | 0.6860 | 0.7459 | 0.7468 |
| GA-S2 | 0.7314 | 0.7514 | 0.7514 | 0.7595 | 0.6025 | 0.7559 | 0.7822 | 0.8004 |
| AVG-S2 | 0.6588 | 0.6806 | 0.7241 | 0.7060 | 0.5880 | 0.7532 | 0.7196 | 0.7641 |

Table 7: FDA performance for various layer selection strategies across different softmax ratios, for GDN-based hybrid students.

| Selector | Llama-3.2-3B-Instruct | | | | Qwen2.5-3B-Instruct | | | |
|---|---|---|---|---|---|---|---|---|
| | 12.5% | 25% | 33% | 50% | 12.5% | 25% | 33% | 50% |
| *Heuristic-Based* | | | | | | | | |
| UNIFORM | 0.7615 | 0.7678 | 0.8236 | 0.8479 | 0.6760 | 0.8515 | 0.8380 | 0.8740 |
| KV | 0.4671 | 0.7894 | 0.8110 | 0.8515 | 0.5311 | 0.8074 | 0.8101 | 0.8272 |
| AR | 0.6850 | 0.8245 | 0.8326 | 0.8263 | 0.7498 | 0.7777 | 0.7984 | 0.8839 |
| VT | 0.4761 | 0.6688 | 0.6895 | 0.8569 | 0.5572 | 0.7255 | 0.7507 | 0.8587 |
| CWE | 0.5878 | 0.6598 | 0.8290 | 0.8569 | 0.5302 | 0.7192 | 0.8020 | 0.8956 |
| ACT-MSE | 0.6121 | 0.7687 | 0.8254 | 0.8380 | 0.5725 | 0.6742 | 0.7939 | 0.8398 |
| LM-PPL | 0.6985 | 0.7714 | 0.7795 | 0.8443 | 0.6535 | 0.7381 | 0.8155 | 0.8353 |
| SMART | 0.7669 | 0.8155 | 0.8074 | 0.8659 | 0.6733 | 0.8335 | 0.8767 | 0.8902 |
| AR-MH | 0.7678 | 0.8254 | 0.8335 | 0.8551 | 0.6445 | 0.7921 | 0.7930 | 0.8299 |
| *Learning-Based (S1 - MSE)* | | | | | | | | |
| GR-S1 | 0.5779 | 0.6958 | 0.7480 | 0.8254 | 0.6688 | 0.7831 | 0.8326 | 0.8821 |
| GA-S1 | 0.5707 | 0.7282 | 0.8146 | 0.8344 | 0.6553 | 0.8047 | 0.8569 | 0.8668 |
| AVG-S1 | 0.5671 | 0.7192 | 0.7957 | 0.8254 | 0.6670 | 0.7975 | 0.8506 | 0.8866 |
| *Learning-Based (S2 - KL)* | | | | | | | | |
| GR-S2 | 0.6301 | 0.8110 | 0.8245 | 0.8425 | 0.8299 | 0.8875 | 0.8749 | 0.8929 |
| GA-S2 | 0.7885 | 0.8506 | 0.8614 | 0.8587 | 0.8398 | 0.8839 | 0.8929 | 0.8992 |
| AVG-S2 | 0.7885 | 0.8137 | 0.8565 | 0.8704 | 0.8128 | 0.8848 | 0.9001 | 0.9109 |

Table 8: SWDE performance for various layer selection strategies across different softmax ratios, for GDN-based hybrid students.

| Selector | Llama-3.2-3B-Instruct | | | | Qwen2.5-3B-Instruct | | | |
|---|---|---|---|---|---|---|---|---|
| | 12.5% | 25% | 33% | 50% | 12.5% | 25% | 33% | 50% |
| *Heuristic-Based* | | | | | | | | |
| UNIFORM | 18.1553 | 21.5138 | 22.1316 | 24.1098 | 10.3260 | 13.2595 | 11.3725 | 16.9006 |
| KV | 17.4030 | 25.5568 | 26.3946 | 29.5483 | 6.6478 | 10.8318 | 16.4796 | 15.2550 |
| AR | 19.5522 | 25.7810 | 28.0807 | 30.8078 | 10.1869 | 12.3800 | 10.1677 | 9.3073 |
| VT | 19.0819 | 24.3118 | 23.9263 | 29.9387 | 7.1499 | 8.7797 | 14.3150 | 18.8876 |
| CWE | 23.7679 | 23.2527 | 28.0014 | 30.3961 | 6.7367 | 12.9678 | 9.9249 | 7.2352 |
| ACT-MSE | 17.2452 | 22.1599 | 24.1714 | 24.8185 | 9.9016 | 8.5959 | 8.8570 | 13.7590 |
| LM-PPL | 20.3075 | 22.2478 | 22.4090 | 27.4774 | 11.1046 | 10.4351 | 9.2945 | 8.1057 |
| SMART | 18.9845 | 28.1036 | 22.8017 | 31.0010 | 14.1967 | 12.4760 | 12.8118 | 16.8912 |
| AR-MH | 24.0994 | 27.0128 | 28.0295 | 29.2224 | 13.7095 | 11.9901 | 11.2729 | 15.8943 |
| *Learning-Based (S1 - MSE)* | | | | | | | | |
| GR-S1 | 13.3918 | 20.7552 | 23.2197 | 27.3407 | 7.8245 | 7.0497 | 9.4220 | 8.6667 |
| GA-S1 | 13.6481 | 17.8867 | 22.6633 | 29.2390 | 8.9412 | 9.0555 | 11.1751 | 9.1234 |
| AVG-S1 | 15.0889 | 18.4342 | 24.3658 | 28.2178 | 7.6409 | 10.6217 | 10.1589 | 10.3181 |
| *Learning-Based (S2 - KL)* | | | | | | | | |
| GR-S2 | 18.0648 | 25.7848 | 30.4299 | 30.5907 | 12.1582 | 6.4855 | 7.8482 | 6.9539 |
| GA-S2 | 25.1144 | 30.9408 | 31.4913 | 33.3520 | 9.7717 | 13.4097 | 13.9486 | 13.9875 |
| AVG-S2 | 23.5556 | 29.2189 | 31.1063 | 32.1499 | 10.6181 | 6.4121 | 6.5623 | 11.3837 |

Table 9: SQuADv2 (F1) performance for various layer selection strategies across different softmax ratios, for GDN-based hybrid students.

## B    Elaboration on Early Stopping for Efficient Selection

**Protocol.**   We study the sample efficiency of our one-swap selector (§3.2) at a fixed hybrid ratio of 1:3 ($K{=}9$ for Qwen2.5-3B-Instruct; $K{=}7$ for Llama-3.2-3B-Instruct). During Stage-2 we train for 4,550 steps and, every 50 steps, compute the current top-$K$ set of layers (from the one-swap importance scores). This yields 91 snapshot sets per model. To quantify stability we analyze each *rolling window* of the last $R{=}10$ snapshots using two complementary views:

- **Rolling pairwise similarity:** the mean pairwise Jaccard over the $R$ sets.

- **Rolling backbone size:** the size of the intersection across the $R$ sets (how many positions are "locked in").

We also relate snapshots to the final selection by reporting the fraction that are *within one swap* of the final consensus (Jaccard $\geq \frac{K-1}{K+1}$; i.e., 0.80 for $K{=}9$ and 0.75 for $K{=}7$).[4]

**Reliable selections emerge well before 4550 steps.**    Two patterns are consistent across both teachers:

- **Qwen2.5-3B-Instruct (K=9).** The run-best set first appears by step 850. From step 1500 onward, 95% of snapshot sets are within one swap of the final consensus; the 10-snapshot rolling Jaccard is high on average ($\approx 0.95$), and rises to 0.99 beyond step 2350.

  By step 1900, the last $R$ snapshots share an 8/9 backbone with at most two candidates for the remaining slot; any one-swap variant at this point attains RULER within 0.007–0.009 absolute points of the run-best (0.8662 vs. 0.8592/0.8582/0.8574).

- **Llama-3.2-3B-Instruct (K=7).** A 6/7 backbone appears by step 750 (mean window Jaccard $\approx 0.91$). The near-optimal set that differs by a single layer first appears at step 1200; from step 1200 onward, 100% of snapshots are within one swap of the final consensus. Stopping here gives RULER 0.6971, within 0.004 absolute of the run-best 0.7011 and comparable to the best late-appearing set.

These observations (i) The selector's rankings stabilize far earlier than the full 4500-step budget; (ii) once the windowed sets agree on $K-1$ layers, the remaining degree of freedom is small and can be resolved cheaply; (iii) one-swap neighbors of the eventual best set typically match downstream RULER within 0.1–1.0 absolute points, so stopping once the $K-1$ backbone is stable is a sound efficiency–quality trade-off.

A conservative choice (see rule below) would have stopped at $\sim 1900$ steps for Qwen and $\sim 1200$ steps for Llama—consuming 42% and 27% of the 4550-step budget, respectively (i.e., 58–74% fewer tokens for the selection pass).

**Practical recipe (rolling-Jaccard early stop).**   Let $S_t$ be the top-$K$ set at step $t$ and $W_t = \{S_{t-9}, \ldots, S_t\}$. Define

$$\text{Backbone}_t = \bigcap_{S \in W_t} S, \quad \text{JaccardMean}_t = \frac{2}{R(R-1)} \sum_{i<j} \text{Jac}(S_i, S_j).$$

Stop at the first step $t$ satisfying:

1. $\text{JaccardMean}_t \geq 0.90$,

2. $|\text{Backbone}_t| \geq K - 1$, and

3. $|\bigcup_{S \in W_t} S| \leq K + 1$ (at most two options for the remaining slot).

*(Optional)* Stop when (3) first becomes true and $S_t \neq S_{t-1}$ to pick the newer of the two candidates.

---

[4]For fixed set size $K$, replacing one layer yields intersection $K-1$ and union $K+1$, hence Jaccard $(K-1)/(K+1)$.

## C   COMPLETE LAYER IMPORTANCE RANKINGS

For all methods that produce a scalar importance score per layer, we obtain hybrid architectures at
target softmax ratios (12.5%, 25%, 33%, 50%) by taking the top-$K$ most important layers according
to that ranking (with $K$ determined by the ratio and total depth $L$). In this section we report the
*full* importance ranking for each such method. Layer indices are zero-based. Methods such as
POSTNAS and SMART do not provide layerwise importance scores, so they are omitted here.

### C.1   QWEN2.5-3B-INSTRUCT

| Selector | Layer indices (most → least important) |
|---|---|
| KV | [1, 0, 26, 19, 18, 20, 5, 17, 27, 6, 15, 22, 24, 16, 3, 11, 23, 21, 28, 8, 14, 25, 2, 29, 32, 12, 13, 9, 4, 10, 31, 34, 35, 30, 33, 7] |
| AR | [0, 1, 27, 18, 20, 25, 24, 26, 21, 8, 12, 19, 23, 7, 35, 17, 33, 22, 28, 16, 32, 30, 34, 9, 29, 2, 6, 5, 31, 4, 13, 10, 14, 15, 3, 11] |
| VT | [0, 1, 19, 26, 28, 25, 35, 10, 15, 17, 3, 7, 27, 29, 16, 14, 30, 34, 32, 31, 23, 33, 9, 13, 18, 8, 2, 21, 11, 12, 22, 24, 20, 5, 4, 6] |
| CWE | [0, 1, 22, 24, 16, 13, 26, 2, 27, 19, 20, 11, 23, 6, 31, 28, 29, 33, 4, 8, 34, 7, 30, 32, 9, 25, 3, 5, 21, 15, 17, 18, 35, 10, 14, 12] |
| ACT-MSE | [0, 1, 35, 34, 31, 33, 32, 30, 8, 12, 27, 3, 4, 2, 6, 5, 28, 10, 9, 29, 11, 7, 14, 13, 26, 25, 16, 15, 18, 24, 17, 23, 20, 19, 22, 21] |
| LM-PPL | [0, 1, 35, 34, 32, 31, 33, 30, 27, 12, 6, 5, 9, 8, 29, 2, 4, 7, 10, 11, 25, 28, 16, 14, 13, 26, 24, 20, 3, 22, 23, 15, 18, 21, 19, 17] |
| AR-MH | [0, 1, 27, 21, 26, 16, 20, 5, 23, 24, 18, 6, 13, 3, 9, 22, 8, 17, 33, 35, 19, 4, 25, 12, 30, 7, 29, 34, 14, 15, 10, 2, 28, 11, 32, 31] |
| GA-S1 | [33, 32, 34, 31, 35, 28, 29, 27, 21, 22, 19, 30, 24, 16, 23, 26, 12, 17, 18, 20, 14, 25, 10, 3, 11, 6, 13, 7, 9, 15, 0, 4, 8, 2, 5, 1] |
| GR-S1 | [33, 32, 34, 35, 31, 27, 28, 30, 21, 29, 22, 19, 26, 25, 16, 24, 23, 17, 18, 14, 15, 12, 20, 13, 11, 10, 8, 9, 7, 6, 3, 5, 4, 0, 2, 1] |
| AVG-S1 | [33, 32, 34, 31, 35, 28, 27, 29, 21, 30, 22, 19, 16, 24, 26, 23, 17, 25, 18, 12, 14, 20, 10, 11, 13, 15, 3, 6, 7, 9, 8, 0, 4, 5, 2, 1] |
| **GA-S2 (OURS)** | [20, 32, 33, 21, 22, 25, 17, 19, 5, 31, 4, 3, 10, 30, 26, 29, 27, 13, 0, 28, 15, 23, 6, 12, 24, 7, 18, 9, 34, 14, 11, 8, 16, 35, 2, 1] |
| GR-S2 | [21, 33, 19, 27, 0, 32, 17, 22, 20, 25, 23, 18, 24, 15, 29, 12, 26, 31, 16, 3, 10, 13, 14, 28, 30, 5, 7, 8, 11, 4, 35, 6, 9, 2, 34, 1] |
| AVG-S2 | [21, 33, 32, 20, 19, 22, 17, 25, 27, 0, 31, 29, 3, 26, 23, 10, 5, 15, 24, 18, 30, 12, 13, 4, 28, 16, 7, 14, 6, 8, 11, 9, 34, 35, 2, 1] |

Table 10: Complete layer-importance rankings for Qwen2.5-3B-Instruct. Each row lists all
$L = 36$ layers from most to least important.

### C.2   LLAMA-3.2-3B-INSTRUCT

| Selector | Layer indices (most → least important) |
|---|---|
| KV | [0, 7, 5, 4, 8, 11, 14, 2, 1, 3, 6, 23, 10, 20, 26, 17, 22, 9, 24, 21, 25, 18, 16, 19, 12, 13, 27, 15] |
| AR | [0, 16, 11, 14, 7, 5, 9, 13, 2, 12, 1, 8, 27, 26, 10, 6, 24, 15, 3, 20, 18, 19, 17, 21, 25, 4, 23, 22] |
| VT | [0, 5, 4, 11, 3, 12, 10, 1, 2, 17, 9, 13, 15, 16, 18, 23, 8, 14, 21, 24, 20, 25, 26, 22, 6, 27, 19, 7] |
| CWE | [0, 5, 12, 8, 9, 4, 1, 13, 14, 10, 21, 24, 16, 22, 15, 27, 25, 20, 6, 2, 26, 23, 18, 3, 11, 17, 19, 7] |
| ACT-MSE | [0, 1, 27, 24, 25, 2, 26, 4, 15, 23, 19, 21, 3, 18, 20, 14, 16, 5, 22, 17, 13, 6, 7, 12, 11, 10, 8, 9] |
| LM-PPL | [0, 1, 27, 2, 24, 3, 26, 25, 4, 14, 15, 19, 16, 5, 20, 23, 12, 17, 10, 21, 13, 18, 6, 22, 9, 11, 7, 8] |
| AR-MH | [0, 13, 12, 16, 11, 7, 23, 14, 10, 5, 21, 25, 9, 8, 19, 17, 2, 6, 4, 3, 1, 26, 18, 24, 15, 22, 27, 20] |
| GA-S1 | [26, 27, 25, 24, 13, 20, 23, 7, 10, 22, 9, 12, 19, 8, 14, 15, 21, 11, 16, 17, 18, 2, 5, 6, 4, 1, 0, 3] |
| GR-S1 | [26, 27, 24, 25, 23, 12, 22, 13, 14, 21, 10, 19, 11, 15, 9, 20, 8, 7, 16, 18, 17, 6, 5, 4, 3, 2, 1, 0] |
| AVG-S1 | [26, 27, 24, 25, 23, 13, 22, 12, 10, 20, 14, 19, 7, 9, 21, 15, 8, 11, 16, 17, 18, 5, 6, 2, 4, 1, 3, 0] |
| **GA-S2 (OURS)** | [14, 8, 5, 12, 15, 13, 2, 26, 24, 16, 17, 18, 21, 10, 25, 20, 19, 23, 22, 27, 9, 7, 6, 4, 1, 0, 11, 3] |
| GR-S2 | [0, 1, 12, 2, 13, 5, 10, 14, 8, 7, 9, 6, 3, 11, 26, 15, 16, 4, 22, 24, 27, 25, 19, 17, 18, 23, 21, 20] |
| AVG-S2 | [12, 5, 14, 2, 8, 13, 10, 15, 26, 0, 1, 16, 24, 7, 9, 6, 17, 18, 25, 22, 19, 21, 3, 11, 27, 4, 20, 23] |

Table 11: Complete layer-importance rankings for `Llama-3.2-3B-Instruct`. Each row lists all $L = 28$ layers from most to least important.

# D LAYER-SELECTION PATTERNS AND SPATIAL ORGANIZATION

We now examine where in depth the selected softmax layers tend to lie, and whether our selector prefers isolated layers or groups of consecutive layers.

**Setup.** For each teacher we take the GA–S2 ranking $\mathcal{R} = (\ell_1, \ldots, \ell_L)$ from Appendix C, ordered from most to least important. For a softmax budget $K$ we define the selected set $S_K = \{\ell_1, \ldots, \ell_K\}$. To quantify how much the selected layers cluster in depth, we use the *adjacency index*

$$A_K = \big| \{ i \in S_K : i + 1 \in S_K \} \big|,$$

i.e., the number of pairs of consecutive layers that are both selected. For a uniformly random $K$-subset of $\{0, \ldots, L - 1\}$, the expected value is $\mathbb{E}[A_K] \approx K(K - 1)/L$, so values substantially above this baseline indicate more clustering than would be obtained by chance. Figure 6 shows the selected indices across budgets, and Figure 7 compares observed and expected adjacency counts.

**Results and discussion.** For **Qwen2.5-3B-Instruct** ($L$=36), GA–S2 produces selected sets that are visibly concentrated in a few depth ranges. At a 25% budget ($K$=9), we obtain $A_K = 4.0$ versus a random baseline of 2.0; at 33% ($K$=12), $A_K = 7.0$ versus 3.68; and at 50% ($K$=18), $A_K = 11.0$ versus 8.49. The plot in Figure 6 show that several of these adjacent pairs occur repeatedly around layers roughly 3–5, 19–22, and 31–33, while the remaining layers are used more sparsely. Thus, the selector does not simply spread the softmax layers uniformly but repeatedly reuses a small number of depth regions as the budget increases.

For **Llama-3.2-3B-Instruct** ($L$=28), the effect is weaker but still present. At 25% ($K$=7), $A_K = 3.0$ versus a baseline of 1.50; at 33% ($K$=9), $A_K = 3.0$ versus 2.58; and at 50% ($K$=14), $A_K = 6.0$ versus 6.50. The selected layers tend to form one main group in the middle of the network (around layers 12–18), with a smaller number of layers near the input and output.

Overall, both models show some degree of clustering beyond what would be expected from a random $K$-subset, but the pattern (multiple groups versus a single main group) depends on the teacher architecture.

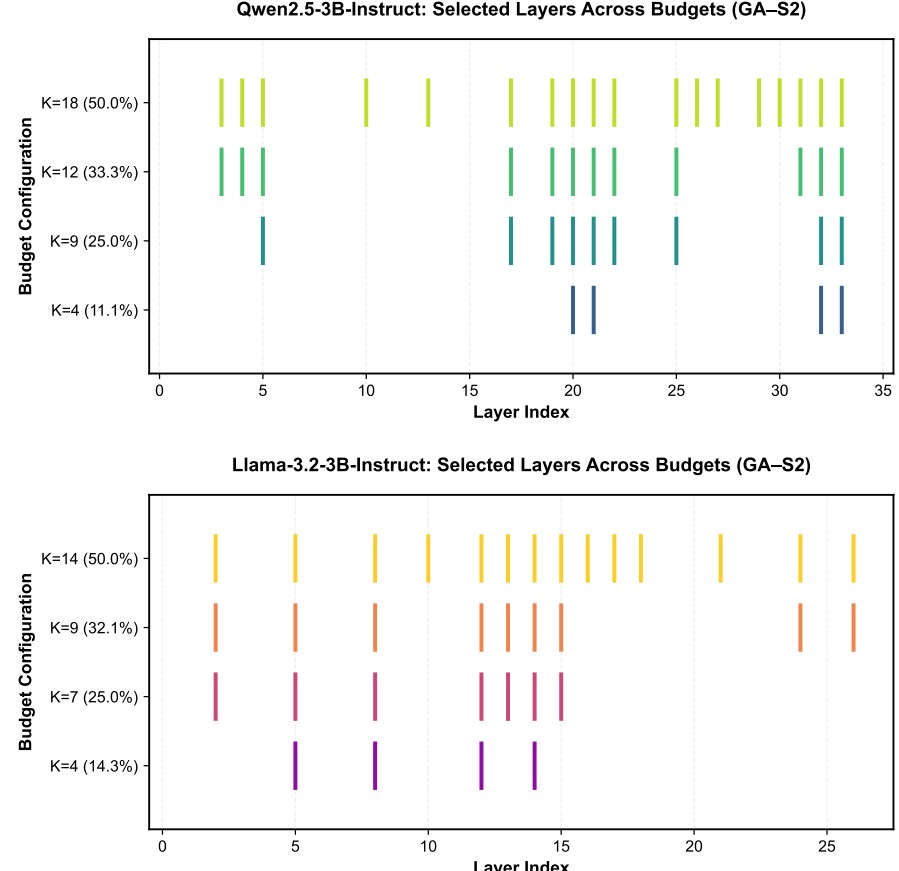

Figure 6: Visualization of selected layers for Qwen2.5-3B-Instruct (top) and Llama-3.2-3B-Instruct (bottom) across budgets (12.5%, 25%, 33%, 50%). Each vertical tick marks a selected layer index.

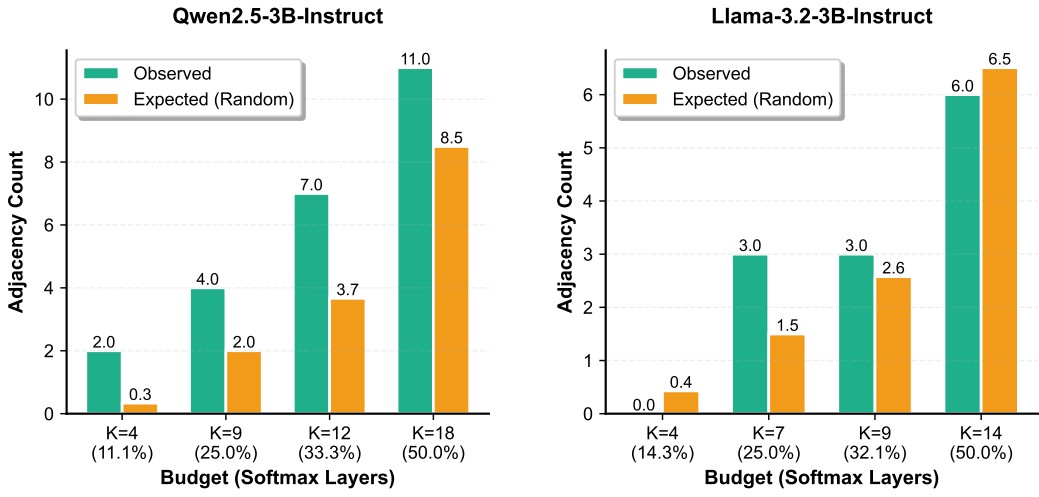

Figure 7: Observed (solid) vs. random-baseline expected (dashed) adjacency counts $A_K$ for Qwen2.5-3B (left) and Llama-3.2-3B (right).

# E    DISTANCE-REGULARIZED SELECTION (DIVERSIFICATION ABLATION)

To probe whether clustering is redundant, we evaluate a re-weighted greedy rule for selecting $K$ layers:

$$\tilde{\mathcal{I}}(\ell \mid S) = \mathcal{I}(\ell) - \lambda \sum_{j \in S} \exp\left(-\frac{|\ell - j|}{\sigma}\right),$$

with $\lambda > 0$, $\sigma > 0$. Here $S$ is the set of softmax layers selected so far and $\mathcal{I}(\ell)$ is the original GA–S2 importance score. The exponential term penalizes placing a new softmax layer too close (in depth) to previously selected ones, nudging the selector toward more spatially diverse configurations without discarding the model-intrinsic KL signal.

We instantiate this diversification for Qwen2.5-3B-Instruct with a GDN student at a fixed 25% softmax ratio ($K{=}9$), and sweep $\lambda \in \{0.025, 0.05\}$ and $\sigma \in \{1, 2\}$. All other training and evaluation settings are kept identical to the main GA–S2 runs.

| $\lambda$ | $\sigma$ | RULER (4096) | **Selected layers** |
|---|---|---|---|
| 0  (GA–S2) | – | 0.8713 | [20, 32, 33, 21, 22, 25, 17, 19, 5] |
| 0.025 | 1 | 0.8509 | [20, 32, 25, 17, 22, 5, 33, 10, 3] |
| 0.025 | 2 | 0.8244 | [20, 32, 25, 5, 17, 10, 33, 0, 22] |
| 0.050 | 1 | 0.8334 | [20, 32, 25, 17, 5, 10, 22, 0, 29] |
| 0.050 | 2 | 0.8303 | [20, 32, 5, 25, 10, 17, 0, 33, 13] |

Table 12: Distance-regularized GA–S2 selection on Qwen2.5-3B-Instruct with a GDN student at a 25% softmax ratio. The $\lambda{=}0$ row corresponds to our default GA–S2 selector without regularization; the last column lists the resulting softmax layer indices.

As shown in Table 12, none of the distance-regularized variants outperform the unregularized GA–S2 selector. A mild penalty ($\lambda{=}0.025$, $\sigma{=}1$) yields a small degradation (0.8509 vs. 0.8713 on RULER), while stronger or more broadly supported penalties lead to larger drops. This suggests that the clustering observed in our selections is not merely redundant: forcing softmax layers to spread out in depth tends to remove genuinely useful local groupings. At the same time, the $\lambda{=}0.025$, $\sigma{=}1$ configuration may be acceptable when a slightly more uniform spatial allocation is desired and a modest recall loss (about two points on RULER) is tolerable.

## F    EXTENDED LONG-CONTEXT EVALUATION VIA NEEDLE-IN-A-HAYSTACK

In the main text, long-context behavior is evaluated primarily through RULER and SWDE (§4, §4.1), whose contexts are below 10k tokens, and our distillation pipeline (§3.1) is trained on generic text with comparatively shorter sequence lengths. This leaves open whether the distilled hybrid model recovers teacher-like retrieval ability at substantially longer sequences than those used during distillation and benchmark evaluation. To probe this, we perform an additional needle-in-a-haystack (NiHA) experiment.

We consider the Qwen2.5-3B-Instruct teacher and its corresponding hybrid student with a 25% softmax / 75% GDN configuration selected by our method. For each context length, we construct inputs by embedding a single target "needle" span into a long filler context and measure retrieval accuracy, defined as the fraction of cases where the model correctly identifies the target span. We evaluate across exponentially increasing context window sizes from 8k to 128k tokens. Results are reported in Table 13.

| Context length (tokens) | Teacher | Hybrid student |
|---|---|---|
| 8,192 | 1.000 | 1.000 |
| 16,384 | 1.000 | 0.998 |
| 32,768 | 1.000 | 0.998 |
| 65,536 | 1.000 | 0.994 |
| 131,072 | 0.954 | 0.684 |

Table 13: Needle-in-a-haystack retrieval accuracy as a function of context length for Qwen2.5-3B-Instruct (teacher) and the corresponding hybrid student (25% softmax, 75% GDN layers).

The hybrid model maintains near-perfect retrieval accuracy up to 65,536 tokens, closely tracking the teacher with only minor degradation. At 131,072 tokens both models begin to degrade, with a larger drop for the hybrid student. These results indicate that the proposed layer selection and distillation procedure successfully preserves long-context retrieval well beyond the context lengths used during distillation and primary benchmark evaluations, while leaving further improvements at extreme lengths as an interesting direction for future work.

# G   ADDITIONAL SCALING RESULTS FOR QWEN2.5 TEACHERS

To verify that our KL-guided layer selection method scales across model sizes within a family, we also distill GDN-based hybrid students from two additional Qwen2.5 teachers:

- **Qwen2.5-1.5B-Instruct**, with RULER score $0.8742$.
- **Qwen2.5-7B-Instruct**, with RULER score $0.9445$.

We use the same DCLM mixture and distillation pipeline as in the main Qwen2.5-3B experiments, and evaluate at 25% and 33% softmax ratios. As in the main text, we compare against UNIFORM, AR, AR-MH, ACT-MSE, LM-PPL, and SMART. Our selector GA-S2 remains consistently stronger than all baselines, particularly in the low-budget regime.

| Model / Ratio | UNIFORM | AR | AR-MH | ACT-MSE | LM-PPL | SMART | **GA-S2** |
|---|---|---|---|---|---|---|---|
| **Qwen2.5-1.5B-Instruct** | (teacher RULER: 0.8742) | | | | | | |
| 25% | 0.4778 | 0.5096 | 0.4243 | 0.3807 | 0.4271 | 0.5098 | **0.5408** |
| 33% | 0.5651 | 0.5552 | 0.5229 | 0.4374 | 0.5056 | 0.6479 | **0.6953** |
| **Qwen2.5-7B-Instruct** | (teacher RULER: 0.9445) | | | | | | |
| 25% | 0.7357 | 0.7453 | 0.7322 | 0.6469 | 0.6544 | 0.8158 | **0.8584** |
| 33% | 0.7516 | 0.8423 | 0.8533 | 0.7227 | 0.6590 | 0.8949 | **0.9110** |

Table 14: RULER performance of GDN-based hybrid students distilled from smaller (1.5B) and larger (7B) Qwen2.5 teachers at 25% and 33% softmax ratios. Our GA-S2 selector consistently outperforms all baselines across scales.

