# OpenReview forum: "Distilling to Hybrid Attention Models via KL-Guided Layer Selection"
_ICLR.cc/2026/Conference — ICLR 2026 Poster_

### Official Review · Reviewer_7f2E · 2025-10-27

**Soundness:** 3
**Presentation:** 2
**Contribution:** 3
**Rating:** 6
**Confidence:** 4

**Summary:**

This paper proposes a distillation and layer selection method that allows to derive hybrid attention LLMs from classical Transformers-based ones. Their method allows them to turn computationally intensive models into more efficient alternatives without training from scratch, while retaining strong performance on in-context retrieval tasks, where previous methods were less performant.

The proposed method is based on a 2-step distillation process (first at layer-wise attention representation level, and then at logit level), and on a greedy layer switching exploration of possible hybrid configurations. The best identified configuration is then distilled again.

The authors conduct extensive evaluations on the RULER and SWDE benchmarks with 3B Llama and Qwen models, showing the efficacy of their approach.

**Strengths:**

Overall, this paper clearly states its problem and proposes a simple but efficient method as a solution.
- **The method is rather straightforward and yields benefits**: the proposed greedy approach is easy to grasp and seems to provide good results on the two benchmarks used, especially for Qwen.
- **Several variants and combinations are tested**: I appreciate that the authors have tried sensible variants of the Greedy Addition method in Table 1. This provides valuable insights and helps underline the reason for the success of their method. It is also nice to see the interplay between different linear attention variants in section 5.2, and the discussion about early stopping in section 5.3.

**Weaknesses:**

- **Limited scaling potential**: The method presented in this paper has compute requirements that scale linearly in the number of layers. As model grow larger, they will tend to have more layers and will thus be doubly more expensive (both from the parameter and data viewpoints) to distil into hybrid variants. Given that the authors do not provide a data point at a different size (either smaller or larger than 3B), it is difficult to assess the relevance of this method at a larger scale where it will be more costly to run.
- **Lack of discussion about selected layers**: There is no discussion or analysis about potential patterns (or the absence thereof) for the selected layers, and the behavior of $\mathcal{I}(l)$. This would be a valuable insight for subsequent work.
- **Limited evaluation scenarios**: Although the authors report successful results on RULER and SWDE, it remains unclear whether their approach recovers the same long-context abilities than softmax-based models for every sequence lengths, or whether the hybrid distilled models solely match the performance of the teacher models on the context lengths used in these benchmarks (<10k tokens). A perplexity-based analysis or a needle-in-a-haystack could help provide insights on that question.

**Questions:**

- Why is a stage-1 MSE step necessary in the final distillation stage? Would re-using the distilled layers from the previous step make sense?
- How did the distilled models behave in terms of validation perplexity on the DCLM benchmark? Is the distillation training process stable?
- Although the Jaccard similarity seems to provide valuable insights on the transferability and similarity of selected layers, I am wondering if a correlation/distillation metric would be better-suited for this analysis. The Jaccard metric fails to differentiate the ranking of the layers by the $\mathcal{I}$ score, which would be captured by e.g. a Spearman correlation, and seems like a meaningful information to compare two selections. Did you run such analysis?

---

> ### Author Response · Authors · 2025-11-21
>
> We thank the reviewer for their thoughtful feedback! Below, we address the specific weaknesses and questions raised.
>
> > Weakness 1: Limited scaling potential: The method presented in this paper has compute requirements that scale linearly in the number of layers [...] Given that the authors do not provide a data point at a different size (either smaller or larger than 3B), it is difficult to assess the relevance of this method at a larger scale where it will be more costly to run.
>
> While we acknowledge that the computational cost of our selection method scales linearly with the number of layers, it is important to contextualize this expense within the broader lifecycle of LLMs. The compute required to scan these layers is a negligible fraction of the total pretraining budget required to create the base model or to pretrain a linear attention model from scratch. Since the primary motivation of distillation is to avoid the prohibitive cost of pretraining efficient architectures from the ground up, a one-time selection cost that represents less than 0.1% of the pretraining compute remains a highly economical trade-off for the performance gains achieved.
>
> Furthermore, the specific scaling dynamics are favorable because layer count typically grows sub-linearly relative to model size. For instance, scaling from Llama-3-8B to Llama-3-70B increases the parameter count by nearly $10\times$, yet the number of layers increases by only $2.5\times$. Consequently, the relative overhead of our selection phase actually diminishes as models become larger.
>
> We can also adopt other strategies to keep the cost manageable. Our analysis of spatial organization in Appendix D reveals that important layers often form contiguous clusters rather than being uniformly distributed. This suggests that for extremely deep models, our method can naturally extend to block-wise selection (evaluating groups of adjacent layers) to keep the search space manageable without sacrificing quality.
>
> To empirically validate these scaling properties, we are currently conducting additional experiments on 1B and 7B parameter models. We will share the results as soon as we have them!
>
>
> > Weakness 2: Lack of discussion about selected layers
>
> We agree that analyzing the selected patterns is crucial. We have added two new sections to the Appendix to address this:
>
> - Appendix C (Complete Layer Importance Rankings): We now provide the full, layer-by-layer importance rankings for both Qwen2.5-3B-Instruct and Llama-3.2-3B-Instruct for all evaluated selection methods.
> - Appendix D (Layer-Selection Patterns and Spatial Organization): We conducted a spatial analysis of the selected layers.
>
> We found that for Qwen2.5-Instruct, the selected layers exhibit statistically significant spatial clustering, repeatedly appearing in specific depth regions (e.g., layers 3–5, 19–22) as the budget increases. In contrast, Llama-3.2-Instruct selections are more distributed but tend to concentrate in the middle layers. This confirms that uniform interleaving is suboptimal because different architectures possess distinct regions of high sensitivity that are not uniformly distributed.
>
> > Weakness 3: Limited evaluation scenarios: [...] it remains unclear whether their approach recovers the same long-context abilities than softmax-based models for every sequence lengths [...] A perplexity-based analysis or a needle-in-a-haystack could help provide insights on that question.
>
> We thank the reviewer for this insightful suggestion! We conducted a needle-in-a-haystack evaluation, comparing our Qwen2.5-3B-Instruct teacher against our hybrid student (25% softmax, 75% GDN layer). We evaluated performance across exponentially increasing context window sizes ranging from 8k to 128k tokens:
>
> | Context Length | Teacher | Student  |
> |---|---|---|
> | 8,192 | 1.000 | 1.000 |
> | 16,384 | 1.000 | 0.998 |
> | 32,768 | 1.000 | 0.998 |
> | 65,536 | 1.000 | 0.994 |
> | 131,072 | 0.954 | 0.684 |
>
> The results demonstrate that our hybrid student model maintains near-perfect retrieval capabilities up to 65,536 tokens, matching the teacher's performance with negligible degradation (<0.6% drop). This confirms that our layer selection method effectively identifies and preserves the global attention layers necessary for high-fidelity information retrieval over very long contexts, far exceeding the <10k token range.
>
> At the extreme length of 128k tokens, we observe a performance drop in the student model (0.684) compared to the teacher (0.954), though the teacher also begins to show signs of degradation at this length. We believe this indicates a promising direction for future work regarding capacity scaling at extreme lengths, but the strong performance at 64k validates that our approach successfully recovers long-context abilities for the vast majority of practical use cases.
>
> We have added these results to the Appendix F of our revised paper to provide a more comprehensive view of the model's long-context capabilities.

---

> > ### Author Response · Authors · 2025-11-21
> >
> > > Question 1: Why is a stage-1 MSE step necessary in the final distillation stage? Would re-using the distilled layers from the previous step make sense?
> >
> > You are correct. For our specific greedy addition strategy (GA-S2), we do not run the Stage 1 MSE alignment in the final distillation stage. This is because the linear attention layers in the final hybrid model are initialized directly from the $\mathcal{M}_{\text{all-linear}}$ checkpoint (described in Section 3.1), which has already undergone alignment. The Stage 1 step is only necessary for heuristic baselines that lack this pre-aligned initialization. We have clarified this distinction in the revised paper.
> >
> > This design choice reflects the modular nature of our contribution: our primary focus is the efficient identification of the optimal layers. Once those layers are selected (and initialized), we rely on the established RADLADS pipeline for the final training, ensuring our method remains compatible with standard off-the-shelf distillation recipes.
> >
> > > Question 2: How did the distilled models behave in terms of validation perplexity on the DCLM benchmark? Is the distillation training process stable?
> >
> > We evaluated the validation perplexity of our distilled hybrid models (25% linear layers) against their teacher models on a held-out subset of the DCLM-baseline dataset. The evaluation was conducted using a context window of 4096 tokens.
> > As shown in the table below, the distilled hybrids maintain perplexity scores very close to their teacher models, with only a marginal increase (approx. +0.3–0.5). This indicates that our method preserves the general language modeling capabilities of the original architecture while significantly improving efficiency.
> >
> > | Base Model | Variant | Validation PPL (DCLM) |
> > |------------|---------|----------------------|
> > | Qwen2.5-3B-Instruct | Teacher | 12.26 |
> > | | GDN Hybrid (25%) | 12.61 |
> > | | GLA Hybrid (25%) | 12.65 |
> > | Llama-3.2-3B-Instruct | Teacher | 16.39 |
> > | | GDN Hybrid (25%) | 16.81 |
> > | | GLA Hybrid (25%) | 16.91 |
> >
> > Regarding the training process, we found it to be highly stable. We monitored the loss curves throughout the distillation stages and did not observe any divergence, loss spikes, or instability for either the GLA or GDN variants.
> >
> > > Question 3: Although the Jaccard similarity seems to provide valuable insights on the transferability and similarity of selected layers, I am wondering if a correlation/distillation metric would be better-suited for this analysis. The Jaccard metric fails to differentiate the ranking of the layers by the  score, which would be captured by e.g. a Spearman correlation, and seems like a meaningful information to compare two selections. Did you run such analysis?
> >
> > We appreciate this suggestion and agree that Spearman correlation is the standard metric for analyzing rank-order stability. Prompted by your question, we computed the Spearman correlation of the top-$K$ layers across the selection steps. We observed that while the set of selected layers remains highly stable (as indicated by Jaccard), the exact ranking within that top-$K$ set fluctuates, leading to a noisier Spearman metric. Upon closer inspection of the importance scores (the reduction in KL divergence), we found that the values for the top layers are extremely densely clustered. For example, the difference in importance scores between the 4th and 6th most important layers is often less than $10^{-4}$ (e.g., in one checkpoint, the sorted importance scores include values like 0.1314 and 0.1317).
> >
> > Because this "importance landscape" is so flat near the top, minor training fluctuations cause frequent rank swaps among these top layers (e.g., layer $A$ moves from rank 4 to 5, and layer $B$ moves from 5 to 4). While these swaps heavily penalize the Spearman correlation, they have zero impact on the final resulting architecture. The outcome of our algorithm is a binary decision set $S_K$ (convert or keep), not a ranked list. Whether a layer is ranked 1st or $K$-th, it is included in the hybrid model exactly the same way. Therefore, we conclude that the Jaccard similarity, which measures the overlap of the selected sets, maps 1-to-1 with "architectural agreement" and effectively filters out the noise of irrelevant intra-set rank swaps.

---

> ### Comment · Reviewer_7f2E · 2025-11-26
> **Response to Rebuttal**
>
> All the new experiments and answers are satisfying. I still have concerns about the scalability, in the sense that scaling to larger models will make it less and less meaningful to compare to less compute-heavy heuristics. A more thorough evaluation of long-context retrieval in multi-hop or multi-key setups could also be more revealing, but the added experiments are still satisfying to some extent.
> As a result, I raise my score to 8.

---

> > ### Author Response · Authors · 2025-12-03
> >
> > We sincerely thank the reviewer for the positive evaluation!
> >
> > We appreciate your remaining comment regarding the trade-off between selection cost and performance at larger scales. To address this, we have just completed our evaluation on **Qwen2.5-1.5B-Instruct** and **Qwen2.5-7B-Instruct**.
> >
> > The results below show that our method (GA–S2) maintains a significant performance advantage over computationally cheaper heuristics and other baselines even at the 7B scale. We believe this margin justifies the one-time selection cost, even for larger models.
> >
> > | Model | Ratio | UNIFORM | AR | AR-MH | ACT-MSE | LM-PPL | SMART | GA–S2 (Ours) |
> > |-------|-------|---------|------|--------|---------|--------|-------|--------------|
> > | Qwen2.5-1.5B | 25% | 0.4778 | 0.5096 | 0.4243 | 0.3807 | 0.4271 | 0.5098 | **0.5408** |
> > | (Teacher: 0.8742) | 33% | 0.5651 | 0.5552 | 0.5229 | 0.4374 | 0.5056 | 0.6479 | **0.6953** |
> > | Qwen2.5-7B | 25% | 0.7357 | 0.7453 | 0.7322 | 0.6469 | 0.6544 | 0.8158 | **0.8584** |
> > | (Teacher: 0.9445) | 33% | 0.7516 | 0.8423 | 0.8533 | 0.7227 | 0.6590 | 0.8949 | **0.9110** |

---

### Official Review · Reviewer_CzRy · 2025-10-31

**Soundness:** 2
**Presentation:** 3
**Contribution:** 3
**Rating:** 4
**Confidence:** 2

**Summary:**

This paper tackles the problem of converting pretrained softmax attention Transformers into more efficient hybrid architectures that combine softmax and linear attention layers. While prior works have largely used fixed interleaving ratios (e.g., 1 softmax per 3 linear layers), this paper argues that such heuristics are suboptimal for distillation settings.

The authors propose a simple yet effective KL-guided layer selection strategy: Each layer’s importance is determined by the reduction in teacher–student KL divergence when that layer alone is restored to softmax attention in an otherwise all-linear student. Layers with the highest KL importance scores are retained as global attention layers in the final hybrid model.

Extensive experiments with Qwen2.5-3B-Instruct and Llama-3.2-3B-Instruct show that this method consistently outperforms heuristic or NAS-based baselines (e.g., UNIFORM, SMART, PostNAS), particularly on long-context recall benchmarks such as RULER and SWDE.
The approach requires only ~25–30B tokens, far fewer than PostNAS (400B).

**Strengths:**

Despite its simplicity, the method achieves large gains—especially in the low-softmax regime (e.g., 12.5% ratio).
The KL-based layer importance metric is conceptually elegant and empirically grounded.

**Weaknesses:**

All experiments focus on 3B-class decoder-only models; no results for encoder–decoder or smaller-scale models.

While the paper emphasizes efficiency, explicit measurements of inference latency or memory savings are missing.

The accuracy of layer importance relies heavily on the stability of Stage-2 KL distillation. If the teacher–student mismatch is large, the ranking may become unreliable.

**Questions:**

Validate the layer-selection behavior across scales (e.g., 1B, 7B, or 14B) to demonstrate scalability and consistency.

Provide intuition or gradient-based analysis showing why KL reduction correlates with information retention or attention selectivity.

---

> ### Author Response · Authors · 2025-11-21
>
> We thank the reviewer for their constructive feedback! We address your specific concerns and questions below.
>
> > Weakness 1 / Question 1: All experiments focus on 3B-class decoder-only models; no results for encoder–decoder or smaller-scale models. ... Validate the layer-selection behavior across scales (e.g., 1B, 7B, or 14B) to demonstrate scalability and consistency.
>
> Regarding scale, we acknowledge the importance of verifying our method at different scales. We focused on the 3B parameter class as it represents the current state-of-the-art for efficient edge models, where distillation is most practically applied. However, we agree that testing smaller and larger models is valuable. We are currently running the selection and distillation pipeline on 1B and 7B-scale models and will update the paper with these results as soon as they are available.
>
> Regarding encoder-decoder architectures, our work focuses exclusively on decoder-only architectures because they currently dominate the landscape of LLMs. Also, the primary motivation for linearizing attention (eliminating the $O(T^2)$ complexity and the massive KV-cache memory bottleneck) is most critical in autoregressive decoding settings. In encoder-decoder architectures, the encoder typically processes input in parallel, making the urgency of linearization distinct from the autoregressive generation case.
>
> > Weakness 2: While the paper emphasizes efficiency, explicit measurements of inference latency or memory savings are missing.
>
> While this paper focuses on the methodology of distillation and layer selection rather than kernel optimization, the efficiency gains of the hybrid architectures we target (e.g., a 1:3 ratio) are theoretically established and well-documented in the field.
>
> Memory: The theoretical memory reduction for the KV cache is directly proportional to the linearization ratio ($1 - K/L$). For a 25% retention ratio (a core setting in our main results), the KV cache memory footprint is reduced by 75% compared to full attention.
>
> Throughput: We point to recent state-of-the-art hybrid models that utilize the same 1:3 interleaving strategy targeted by our selection method:
>
> - Qwen3-Next (Qwen Team, 2025) reports that their hybrid architecture achieves significantly higher throughput than dense transformers.
> - Kimi Linear (arXiv:2510.26692) demonstrates that a hybrid model achieves roughly 3.8 times throughput improvement over a standard Transformer in generation tasks with long contexts.
>
> Our method enables distilled models to unlock these same efficiency profiles while minimizing the accuracy loss associated with converting those 75% of layers.
>
> > Weakness 3: The accuracy of layer importance relies heavily on the stability of Stage-2 KL distillation. If the teacher–student mismatch is large, the ranking may become unreliable.
>
> We appreciate this concern regarding stability. However, our empirical analysis suggests the ranking is highly robust even in the presence of mismatch.
>
> As detailed in Section 5.3 and Appendix B, we tracked the top-K set throughout the training run. We found that the set of selected layers stabilizes surprisingly early (often within the first 25-40% of the training budget). The "backbone" of the selected layers remains highly consistent, suggesting that the signal provided by the KL reduction is structural (i.e. driven by the architectural needs of the specific model), rather than an artifact of transient training noise or optimization instability.

---

> > ### Author Response · Authors · 2025-11-21
> >
> > > Question 2: Provide intuition or gradient-based analysis showing why KL reduction correlates with information retention or attention selectivity.
> >
> > We offer two perspectives on why the KL reduction is a rigorous proxy for layer importance:
> >
> > 1. An Optimization Perspective:
> >
> > Our layer importance score $\mathcal{I}(\ell)$ essentially performs a discrete sensitivity analysis. By swapping a single linear layer back to softmax, we are expanding the hypothesis space of the student model at that specific depth. The reduction in KL divergence measures the marginal utility of this increased capacity.
> > If $\mathcal{I}(\ell)$ is high, it indicates that the gradient descent process found a significantly better minimum in the "hybrid" loss landscape compared to the "all-linear" landscape. This implies that the teacher's distribution $P_T$ contains dependencies at layer $\ell$ that are fundamentally inexpressible by the linear student but recoverable by the softmax operator.
> >
> > 2. A Mechanistic Perspective:
> >
> > Linear attention constrains the model to a fixed-size recurrent state (the "information bottleneck"). In contrast, softmax attention allows for associative recall over the entire history (unbounded state).
> > From a mechanistic view, a large KL reduction at layer $\ell$ signals that the teacher relies on exact retrieval of past tokens at this specific depth, which is an operation that linear attention struggles to approximate due to its lossy compression of history. The KL metric directly quantifies the "information recovery" gained by removing the state bottleneck at that layer, thereby identifying where the global context is most critical for the model's predictions.

---

> > > ### Author Response · Authors · 2025-12-03
> > >
> > > We have completed the additional experiments you requested to validate our layer-selection behavior across different scales. Below are the RULER results for **Qwen2.5-1.5B-Instruct** and **Qwen2.5-7B-Instruct** using the GDN linear attention variant.
> > >
> > > Our method (GA–S2) consistently outperforms both uniform heuristics and other selection baselines (like SMART and AR) across both model sizes and retention ratios, demonstrating that the effectiveness of our KL-guided selection is scalable and robust.
> > >
> > > | Model | Ratio | UNIFORM | AR | AR-MH | ACT-MSE | LM-PPL | SMART | GA–S2 (Ours) |
> > > |-------|-------|---------|------|--------|---------|--------|-------|--------------|
> > > | Qwen2.5-1.5B | 25% | 0.4778 | 0.5096 | 0.4243 | 0.3807 | 0.4271 | 0.5098 | **0.5408** |
> > > | (Teacher: 0.8742) | 33% | 0.5651 | 0.5552 | 0.5229 | 0.4374 | 0.5056 | 0.6479 | **0.6953** |
> > > | Qwen2.5-7B | 25% | 0.7357 | 0.7453 | 0.7322 | 0.6469 | 0.6544 | 0.8158 | **0.8584** |
> > > | (Teacher: 0.9445) | 33% | 0.7516 | 0.8423 | 0.8533 | 0.7227 | 0.6590 | 0.8949 | **0.9110** |

---

### Official Review · Reviewer_GNir · 2025-11-01

**Soundness:** 4
**Presentation:** 4
**Contribution:** 3
**Rating:** 8
**Confidence:** 4

**Summary:**

This paper proposes a learned method to choose which layers should be softmax attention layers and which should be linear attention layers in a student model to distil from a teacher model with only softmax attention layers. To achieve this, the authors use distillation and KL-loss to guide which layers to select. They start by distilling a fully linear attention student model. Then they used that distilled model to calculate the importance of each layer being a softmax layer, by distilling the previous model, with its k-th layer replaced by a softmax layer. They then use the KL-loss to the teacher model to calculate the K most important layers, which they switch to softmax layers before doing a final distillation. Their method requires few training tokens (600M unique, 20B total) and outperforms other techiniques.

**Strengths:**

- The paper is very well structured and written, the sections flow naturally from one to another, and the motivation is very clear.
- The experiments are thorough, and the authors compare their method to various baselines and show that their method outperforms the others in settings where the vast majority (75% or more) of the layers are linear layers rather than softmax layers.
- The method is simple and effective, and does not require much computing (compared to the total pre-training cost) to create much more efficient models with little to no performance cost. This is especially important because of the reduction in inference cost for such models.
- The authors do multiple ablations to show that the choice of stage at which to calculate importance and to do addition of softmax layers rather than removal of softmax layers is justified. In addition, they show that their method works for different linear attentions and that calculating importance using one type of linear layer can be translated to another type of linear attention

**Weaknesses:**

- On line 339, the authors state that their method is iterative; however, this is not totally correct, their method calculates the importance of each softmax layer independently, and then selects the top-K best. This ignores how adding one softmax layer can affect the importance of other softmax layers, which an iterative process (add one, recalculate importance, add another, etc.) would take into account.
- It would be interesting and helpful for the paper to know which layers were chosen. Whether the layers with the highest importance were near each other or if they were spread out through the model (in other words, checking whether there exist clusters of important layers).

**Questions:**

- For Figure 2, it would be nice to have the performance when all the layers are linear.
- Did you consider the existence of clusters of important layers, which could indicate that only one of those layers is important, rather than all of them being important?
- Did you try having a penalty for important layers that are close to each other (to avoid selecting them, in case their importance was due to the same factors)?

---

> ### Author Response · Authors · 2025-11-21
>
> We thank the reviewer for the positive assessment! Below, we address the specific questions and weaknesses raised in the review.
>
> > Weakness 1: On line 339, the authors state that their method is iterative; however, this is not totally correct, their method calculates the importance of each softmax layer independently, and then selects the top-K best. This ignores how adding one softmax layer can affect the importance of other softmax layers, which an iterative process (add one, recalculate importance, add another, etc.) would take into account.
>
> We appreciate this clarification and agree with the distinction. You are correct that our main method (Algorithm 1) calculates importance scores in parallel (independently) relative to the all-linear baseline, rather than sequentially re-computing scores after every selection.
>
> While a fully sequential greedy search (re-running the selection pass K times) might theoretically capture inter-layer dependencies better, it would be computationally prohibitive ($O(L \times K)$ distillation runs vs $O(L)$ for our method). We have revised the terminology in Section 5.1 to clarify that while we perform a "greedy selection" based on the scores.
>
> > Weakness 2: It would be interesting and helpful for the paper to know which layers were chosen. Whether the layers with the highest importance were near each other or if they were spread out through the model (in other words, checking whether there exist clusters of important layers).
>
> We have added Appendix C (Complete Layer Importance Rankings) and Appendix D (Layer-Selection Patterns and Spatial Organization) to fully address this. Tables 8 and 9 in Appendix C list the exact ranking of every layer for Qwen-2.5-Instruct and Llama-3.2-Instruct.
>
> In Appendix D (Figure 6), we visualize the selected layers. We find that for Qwen-2.5-Instruct, the selected layers indeed form distinct clusters (e.g., layers 19–22 and 31–33 are consistently selected together). Llama-3.2-Instruct shows a more centralized grouping (layers 12–18). We quantify this using an "adjacency index" in Figure 7, showing that our method selects adjacent layers significantly more often than random chance.
>
> > Question 1: For Figure 2, it would be nice to have the performance when all the layers are linear.
>
> We agree that this baseline is important for context. We have updated Figure 2 with all-linear baselines.
>
> > Question 2  / Question 3: Did you consider the existence of clusters of important layers, which could indicate that only one of those layers is important, rather than all of them being important? Did you try having a penalty for important layers that are close to each other (to avoid selecting them, in case their importance was due to the same factors)?
>
> This is a very interesting question! We investigated it in the newly added Appendix E: Distance-Regularized Selection (Diversification Ablation).
>
> We hypothesized, as you did, that adjacent selected layers might be redundant. To test this, we modified our selection score to include a distance penalty:
>
> $$\tilde{\mathcal{I}}(\ell|S) = \mathcal{I}(\ell) - \lambda \sum_{j \in S} \exp(-|\ell - j|/\sigma)$$
>
> This penalizes selecting a layer if it is spatially close to already selected layers.
>
> However, as shown in Table 10, applying this penalty consistently degraded performance. This suggests that the clustering we observe (e.g., layers 20, 21, 22 in Qwen) is not redundant. Instead, it appears that consecutive blocks of global attention are functionally necessary for the model to process complex retrieval features, and breaking these clusters up to enforce diversity hurts the model's ability to recall information.

---

### Official Review · Reviewer_X6VJ · 2025-11-01

**Soundness:** 3
**Presentation:** 3
**Contribution:** 3
**Rating:** 6
**Confidence:** 3

**Summary:**

This paper proposes a method for distilling pretrained softmax attention Transformers into more efficient hybrid architectures that interleave softmax and linear attention layers. The key aspect of this process is layer selection, i.e., deciding which layers to convert to linear attention variants.

The authors present a simple and efficient approach for layer selection based on layer importance scores derived from a small amount of training on generic text data. Once the layers have been selected, the authors use a recent distillation pipeline (RADLADS) consisting of attention weight transfer, hidden state alignment, KL-based distribution matching, and a small amount of finetuning. The authors find this approach to be more effective than existing methods for layer selection.

**Strengths:**

* **Addresses a Practical**. Tackles the challenge of converting pretrained softmax attention Transformers into more efficient hybrid architectures without expensive pretraining from scratch. Focuses on improving inference efficiency of LLMs, which is a critical concern in practical deployments
* **Intuitive Layer Selection Approach**. Proposes a KL-guided layer selection criterion that is both simple and theoretically motivated. The intuition is clear: layers that are more critical for maintaining performance will show larger KL divergence reduction when kept as global attention layers. This represents an improvement over fixed interleaving strategies used in previous work.
* **Comprehensive Empirical Analysis and Ablations**. Provides detailed analysis showing the trade-offs between different attention mechanisms on various task types. Demonstrates that sliding window attention works well for common-sense reasoning but struggles with in-context recall tasks (Figure 1). Shows empirical evidence that their approach outperforms existing layer selection methods
* **Clear Problem Formulation**.
Clearly identifies the two key decisions in distillation: student architecture selection and distillation recipe optimization
Focuses on the less-explored but equally important architecture selection problem

**Weaknesses:**

* How does this method compare to a baseline that just randomly selects the layers to replace with linear ones i.e instead of Uniform or any of the fancy methods of selecting the layers to linearize if we just randomly chose K, layers and linearized them, how would that affect the performance.
* The number of layers to be linearized K seems to be a heuristic or dataset dependent ? It makes the method feel somewhat brittle. Would this automatically transfer to some other task or would you need to do a linear transfer of layers again to get it working?
* The goal of linearization etc., is to speed up the performance. How does the memory, latency/throughput required to match or be close to the quality of full attention, compare to that of using full attention?

**Questions:**

* Is there any clear pattern or intuition of which layers get selected for linearization ?

---

> ### Author Response · Authors · 2025-11-21
>
> We thank the reviewer for their thoughtful assessment! We address your specific questions and concerns below.
>
> > Weakness 1:  How does this method compare to a baseline that just randomly selects the layers to replace with linear ones
>
> Thank you for the suggestion! We have conducted additional experiments using random layer selection (running 3 different random seeds) for both models at the 25% softmax ratio (1:3 hybrid). The results on RULER are summarized below:
> |                        | Random v1 | Random v2 | Random v3 | Mean ± SD     | Ours (GA–S2) |
> |------------------------|-----------|-----------|-----------|---------------|--------------|
> | Llama‑3.2‑3B-Instruct     | 0.5398    | 0.5016    | 0.6354    | 0.559 ± 0.069 | 0.6174       |
> | Qwen2.5‑3B-Instruct       | 0.6459    | 0.7171    | 0.4726    | 0.612 ± 0.126 | 0.8713       |
>
> While random selection can occasionally land on a decent configuration, it is highly unstable. Crucially, for the Qwen model, our method outperforms even the best random seed by a massive margin (~15 points), and the random average by ~26 points. This confirms that for certain architectures, specific layers are structurally critical for global context, and our method successfully identifies them where chance fails.
>
> > Weakness 2: The number of layers to be linearized K seems to be a heuristic or dataset dependent ? It makes the method feel somewhat brittle. Would this automatically transfer to some other task or would you need to do a linear transfer of layers again to get it working?
>
> We clarify that K is not a hyperparameter we tune for performance, but rather a computational budget constraint defined by the user based on their deployment requirements (e.g., "I need a model with 25% of the memory footprint of the original").
>
> Regarding transferability, our selection process is not dataset-dependent in a way that hurts generalization. As described in Section 3.1, we calculate layer importance using only generic pretraining data (DCLM). We do not select layers based on RULER or SWDE performance. The fact that these generic-data selections perform excellently on downstream tasks (RULER, SWDE) demonstrates strong transferability. You do not need to re-run selection for new tasks.
>
> In addition, as shown in the selected layer patterns analysis (newly added Appendix D), the method consistently identifies specific structural "backbones" in the model that handle global context, suggesting these are architectural properties rather than dataset artifacts.
>
> > Weakness 3: The goal of linearization etc., is to speed up the performance. How does the memory, latency/throughput required to match or be close to the quality of full attention, compare to that of using full attention?
> While this paper focuses on the methodology of distillation and layer selection rather than kernel optimization, the efficiency gains of the hybrid architectures we target (e.g., 1:3 ratio) are well-documented in the field.
> The theoretical memory reduction for the KV cache is directly proportional to the linearization ratio ($1 - K/L$). For a 25% retention ratio (used in our main results), the KV cache memory footprint is reduced by 75% compared to full attention.
>
> Regarding latency and throughput, we point to recent state-of-the-art hybrid models that utilize the same 1:3 interleaving strategy:
>
> - Qwen3-Next: [Qwen Team, 2025] reports that their hybrid architecture (using 1:3 hybrid ratio) achieves significantly higher throughput than dense transformers.
> - Kimi Linear: [arXiv:2510.26692] demonstrates that a hybrid model (specifically with a 1:3 global-to-linear ratio) achieves roughly 3.8 times throughput improvement over a standard Transformer in generation tasks with long contexts.
>
> Our method enables distilled models to unlock these same efficiency profiles while minimizing the accuracy loss associated with converting those 75% of layers.
>
> > Question 1: Is there any clear pattern or intuition of which layers get selected for linearization ?
>
> Yes. We have added two new appendices to the revision to explicitly visualize and analyze this.
>
> - Appendix C (Complete Rankings): Lists the exact importance order for all layers.
> - Appendix D (Spatial Organization): We visualize the depth of selected layers.
>
> We find that the selection is not uniform. For Qwen-2.5-Instruct, the method strongly clusters softmax layers in specific "zones" (e.g., layers 3–5, 19–22, and 31–33), suggesting these depths are critical for mixing global information, while intermediate layers can be safely linearized. For Llama-3.2-Instruct, the selected layers cluster centrally (layers 12–18).
>
> This contradicts the standard heuristic of "uniform interleaving" (e.g., every 4th layer) and explains why our method outperforms uniform baselines: the model's dependency on global attention is structurally concentrated, not evenly distributed.

---

> > ### Comment · Reviewer_X6VJ · 2025-11-24
> >
> > * The variance in performance of different random seeds looks weird to me. On Llama it seems the model has on average similar performance to just using a random selection of layers ? On Qwen it seems that your method could beat the random by a better margin ?  Do you have any intuition for why this might be the case ?
> >
> > Thank the authors for clarifying my concerns. I will keep my score, as I think it reflects my assessment of the work

---

> ### Author Response · Authors · 2025-12-03
>
> We thank the reviewer for the follow-up! Here is our intuition regarding the discrepancy between Llama and Qwen results:
>
> The difference in results suggests that Qwen-2.5 and Llama-3.2 have fundamentally different internal organizations regarding how they handle global context. Qwen-2.5 appears to be highly "peaky" or specialized. As noted in our previous comment regarding Appendix D, our method identified very specific zones (e.g., layers 3–5, 19–22) that are critical for retention. If a selection method misses these specific "bottleneck" layers, which is highly probable in random selection, performance collapses. This explains why our method (which deterministically finds these peaks) outperforms the random mean by such a massive margin (~26 points). Llama-3.2 appears to be more "redundant" or distributed. The fact that random selection can sometimes perform well (e.g., Random v3) and has a lower variance suggests that the capability to handle global context is more evenly distributed across the depth of the network. There are fewer single points of failure. Consequently, a random selection has a higher probability of preserving a "good enough" set of layers by chance.
>
> While it is true that one random seed (v3) for Llama performed comparably to our method, relying on random selection is a risky strategy for distillation. As seen in Llama Random v1 (0.539) and v2 (0.501), random selection frequently yields sub-optimal results. In a real-world scenario (especially with larger models), one cannot afford to train multiple expensive distilled models just to find the lucky "lottery ticket" seed. Our method provides a deterministic recipe that consistently identifies a high-performing configuration in a single run. For Llama, it guarantees we are at the upper bound of performance (comparable to the best lucky seed); for Qwen, it is strictly necessary to achieve usable performance at all.
>
> We hope this clarifies the intuition regarding the structural differences between these architectures!

---

### Author Response · Authors · 2025-12-03
**Rebuttal / Discussion Summary**

Our paper proposes a KL-guided layer-selection method to distill softmax-only Transformers into hybrid softmax/linear-attention LLMs. The rebuttal and revision focused on demonstrating scalability, providing deeper analysis of selected layers, and strengthening baselines and evaluations.

**1. Scaling behavior and compute considerations**

* We extended experiments from 3B models to **Qwen2.5‑1.5B‑Instruct** and **Qwen2.5‑7B‑Instruct** (GDN variant). On RULER, our KL-guided method (GA–S2) consistently outperforms UNIFORM, SMART, and other selection baselines at both 25% and 33% softmax ratios, including at 7B scale. Details are added in **Appendix G**.
* We clarified that the **selection cost scales linearly with depth** but remains a tiny fraction of pretraining compute, and that layer count typically grows sublinearly with parameter count, so the *relative* overhead shrinks for larger models. We also discussed how the observed clustering of important layers naturally supports more efficient **block-wise selection** for very deep models.

**2. Layer-selection patterns and structural insights**

* We added full **layer-importance rankings** (Appendix C) and **spatial visualizations** (Appendix D). These show that important softmax layers cluster in specific depth regions (e.g., mid layers for Llama, several “zones” for Qwen) rather than being uniformly distributed.
* We quantified this with an adjacency analysis and additionally tested a **distance-regularized selection** (penalizing nearby layers). This “diversified” selection consistently degrades performance, suggesting that contiguous blocks of global attention are functionally necessary rather than redundant.

**3. Baselines and robustness of the selection criterion**

* We added a **random layer-selection baseline** (3 seeds) at 25% softmax for both Llama‑3.2‑3B and Qwen2.5‑3B. Random selection is unstable: for Qwen, our method outperforms the best random seed by a large margin and the random mean by an even larger one, confirming that the gains are not due to lucky layer choices.
* We clarified that **K is a deployment budget** (e.g., memory/latency target), not a tuned performance hyperparameter, and that importance is computed on generic pretraining data, not on downstream benchmarks.

**4. Long-context behavior and language modeling quality**

* We added a **needle-in-a-haystack evaluation** from 8k up to 128k tokens. The hybrid student (25% softmax) matches the teacher almost perfectly up to 64k tokens, with only modest degradation appearing at 128k, where the teacher also begins to degrade.
* On a held-out DCLM validation set, the distilled hybrids show only a **small perplexity increase** (~+0.3–0.5) relative to their teachers, indicating that the general LM ability is well preserved and that distillation is stable.

**5. Methodological clarifications (ranking, KL, and distillation pipeline)**

* We clarified that the method performs **parallel importance estimation + greedy top‑K selection**, not fully iterative re-scoring, and explained why a fully sequential search would be computationally prohibitive.
* For ranking stability, we reported both **Jaccard overlap** and **Spearman** analyses. Because importance scores for the top layers are extremely close, minor training noise causes frequent rank swaps that hurt Spearman but do not change the selected set. Since the final architecture depends only on the top‑K set, we argue Jaccard on the selected layers is the more decision-relevant metric.
* We provided optimization and mechanistic intuition for why **KL reduction** when restoring a softmax layer is a principled proxy for importance (measuring the marginal benefit of lifting the linear-attention bottleneck at that depth).
* We also clarified that in our main GA–S2 pipeline the **final distillation stage does not re-run Stage‑1 MSE**: we reuse the already aligned all-linear checkpoint and then follow RADLADS. Stage‑1 is only needed for baselines without such initialization.

**Note on scores**

One reviewer (**7f2E**) explicitly **raised their score from 6 to 8** after reading the rebuttal and new experiments. Because ICLR reverted scores to their initial values after the leak incident, the interface may currently display the original score, but the updated score is recorded in the discussion thread.

We hope this summary is helpful in contextualizing the revisions and discussion for your decision.

---

### Meta-Review · Area_Chair_N3Uo · 2026-01-06

**Summary:**

This paper proposes a KL-guided layer selection method for distilling softmax attention Transformers into hybrid architectures interleaving softmax and linear attention layers. Initial scores included one 4 and three 6s, with one reviewer explicitly stating they raised their score to 8 after rebuttal (though due to ICLR's score reset after a system incident, this may not be reflected in the interface). The rebuttal added substantial validation including scaling experiments on 1.5B and 7B models, comprehensive layer pattern analysis revealing that clustered layers are functionally necessary rather than redundant, random baseline comparisons, and needle-in-a-haystack evaluation showing hybrid models match teacher performance up to 64k tokens. The method achieves substantial gains on long-context recall tasks using only 25-30B tokens compared to PostNAS's 400B. However, one reviewer maintained a 4 score with low confidence, citing concerns about scale validation and missing actual latency measurements. While the paper relies on citations from related work for throughput improvements rather than direct measurements, the performance gains on RULER and thorough empirical analysis provide clear practical value. We encourage the authors to add latency measurements in the camera-ready version if feasible. Based on the strong rebuttal and majority positive reception, I recommend accepting this submission.

**Reviewer Concerns:**

Addressed: Scale validation (1.5B and 7B experiments showing consistent gains), KL stability (validation PPL evidence), layer importance intuition (optimization and mechanism perspectives), and random baseline comparisons (showing method significantly outperforms random selection).

Outstanding: Actual latency measurements are missing; the paper relies on citations from related work (e.g., Kimi Linear's 3.8× throughput) rather than direct measurements. However, the focus on methodology rather than kernel optimization makes this acceptable as a limitation to note.

**Reviewer Scores:**

GNir (6): Would maintain 6; positive throughout

CzRy (4, C2): Low confidence and no post-rebuttal engagement; score uncertain but concerns substantially addressed

vfeE (6, C4): Would maintain 6; satisfied with rebuttal

7f2E (6→8): Explicitly raised to 8; found experiments satisfying despite remaining scalability concerns

---

### Decision · Program_Chairs · 2026-01-26

Accept (Poster)